# FROM CONVERSATION TO QUERY EXECUTION: BENCHMARKING USER AND TOOL INTERACTIONS FOR EHR DATABASE AGENTS

**Gyubok Lee**[1]*, **Woosog Chay**[1], **Heeyoung Kwak**[2], **Yeong Hwa Kim**[2]
**Haanju Yoo**[2], **Oksoon Jeong**[3], **Meong Hi Son**[3], **Edward Choi**[1]
KAIST[1] NAVER Cloud[2] Samsung Medical Center[3]
{gyubok.lee,edwardchoi}@kaist.ac.kr[1]

## ABSTRACT

Despite the impressive performance of LLM-powered agents, their adoption for Electronic Health Record (EHR) data access remains limited by the absence of benchmarks that adequately capture real-world clinical data access flows. In practice, two core challenges hinder deployment: *query ambiguity* from vague user questions and *value mismatch* between user terminology and database entries. To address this, we introduce *EHR-ChatQA,* an interactive database question answering benchmark that evaluates the end-to-end workflow of database agents: clarifying user questions, using tools to resolve value mismatches, and generating correct SQL to deliver accurate answers. To cover diverse patterns of query ambiguity and value mismatch, EHR-ChatQA assesses agents in a simulated environment with an LLM-based user across two interaction flows: Incremental Query Refinement (IncreQA), where users add constraints to existing queries, and Adaptive Query Refinement (AdaptQA), where users adjust their search goals mid-conversation. Experiments with state-of-the-art LLMs (e.g., o4-mini and Gemini-2.5-Flash) over five i.i.d. trials show that while the best-performing agents achieve Pass@5 of over 90% (at least one of five trials) on IncreQA and 60–70% on AdaptQA, their Pass$^\wedge$5 (consistent success across all five trials) is substantially lower, with gaps of up to about 60%. These results underscore the need to build agents that are not only performant but also robust for the safety-critical EHR domain. Finally, we provide diagnostic insights into common failure modes to guide future agent development. Our code and data are publicly available at https://github.com/glee4810/EHR-ChatQA.

## 1 INTRODUCTION

Large Language Models (LLMs) are increasingly operating as autonomous agents, interacting with external environments to solve complex tasks. One key application is interfacing with structured databases, which can substantially enhance data accessibility for non-technical users. This capability is particularly impactful in high-stakes domains such as Electronic Health Records (EHRs), where enabling natural language queries over vast patient data repositories has the potential to fundamentally transform both clinical research and patient care (Ohno-Machado, 2011; Yang et al., 2022). To assess such capabilities, the prevailing evaluation paradigm has relied on text-to-SQL benchmarks (Yu et al., 2018; Wang et al., 2020; Lee et al., 2022; Li et al., 2023), which measure a model's ability to translate natural language questions into SQL queries. However, these benchmarks primarily emphasize the isolated task of mapping a single, well-formed question to SQL, a setting that fails to capture the complexities of how clinicians interact with EHRs.

To further investigate the gap between this evaluation paradigm and real-world clinical needs, we collaborated with a major academic medical center and reviewed internal text-to-SQL query logs. This analysis revealed that existing benchmarks are not suitable for capturing two core real-world

---

*This work was mostly done during his internship at NAVER Cloud.

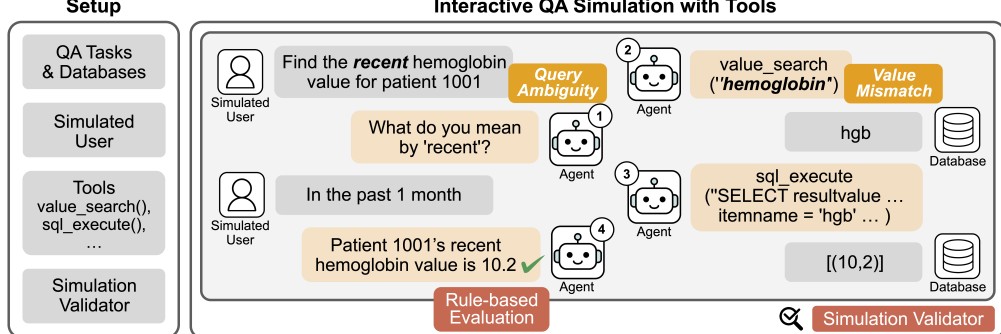

Figure 1: Overview of EHR-ChatQA. Our benchmark places an agent in a simulated environment with an LLM-powered user and tools. The agent must manage the entire workflow autonomously: clarifying vague user questions, resolving terminology mismatches, generating and executing accurate SQL, and returning the final answer. When the interaction is over, each trace is verified by a simulation validator, and only traces that pass proceed to rule-based evaluation.

challenges, which significantly hinders the adoption of these models: (1) *Query Ambiguity*[1]: Users often pose vague questions that do not fully capture their true intent (e.g., a request such as "Show me recent labs" lacks specifics such as test types or time ranges) (Wang et al., 2023; Saparina & Lapata, 2024; Dong et al., 2025). (2) *Value Mismatch*: Clinical terms in EHRs vary due to specialized database naming conventions and often differ from everyday language (e.g., "Lopressor" may not match the database entry "metoprolol tartrate"), creating a terminological gap (Holmes et al., 2021; De Mello et al., 2022) that must be bridged for accurate SQL generation.

Addressing these challenges requires moving beyond static, single-turn SQL generation to an environment where an agent can clarify a user's intent, invoke necessary tools to navigate complex EHR schemas and clinical values, and synthesize all relevant information to generate an accurate SQL query. To bridge this gap, we introduce *EHR-ChatQA*, an interactive database question answering (QA) benchmark designed to assess this end-to-end agentic workflow, from conversation to query execution, in the EHR domain. By placing agents in a simulated environment with both an LLM-based user and a suite of tools, EHR-ChatQA provides a holistic evaluation of agent capabilities in interactive clinical data access flows. Each interaction trace is further verified by a dedicated validator to ensure evaluation reliability. Our benchmark is grounded in real-world clinical QA scenarios and two publicly available EHR databases (MIMIC-IV and eICU). It consists of tasks categorized into two interaction flows designed for various query ambiguities and value mismatches: Incremental Query Refinement (IncreQA), where users add new constraints to a query, and Adaptive Query Refinement (AdaptQA), where users adjust their search goals mid-conversation.

Our evaluation of various state-of-the-art LLMs on EHR-ChatQA reveals a critical lack of robustness under diverse conversation paths. While agents often succeed in at least one of five attempts on a task (Pass@5), their ability to succeed consistently in all five attempts (Pass$^\wedge$5) is substantially lower. This performance gap exceeds 38% for IncreQA and 36% for AdaptQA, reaching up to about 60%. This inconsistency highlights a crucial lack of reliability in current agents, raising significant concerns for their deployment in safety-critical domains such as EHRs and pointing to key areas for future research.

The main contributions of our work are summarized as follows:

- We propose EHR-ChatQA, the first interactive benchmark for EHR QA that holistically evaluates agents' interactive, end-to-end workflows using simulated users and a set of customizable tools for schema exploration, value exploration, and web search.

- Grounded in real-world clinical QA scenarios and two publicly available EHR databases, the benchmark contains two interaction flows to reflect various query ambiguity and value mismatch patterns.

---

[1]While linguistics distinguishes it from vagueness (a lack of specificity), we adopt the common NLP convention of using "ambiguity" to refer to any query requiring clarification.

- Our evaluation of various LLMs reveals a critical performance gap between an agent's optimistic success (Pass@5) and its consistent success (Pass$^\wedge$5), providing diagnostic insights for developing more performant and reliable agents in interactive EHR QA.

## 2 RELATED WORK

**Text-to-SQL Benchmarks** Text-to-SQL research has largely focused on translating a single, well-defined question into an SQL query. Benchmarks such as Spider (Yu et al., 2018) and BIRD (Li et al., 2023) are prominent examples that have shaped this paradigm. While other benchmarks such as SParC (Yu et al., 2019b), CoSQL (Yu et al., 2019a), and PRACTIQ (Dong et al., 2025) introduced conversational context, their evaluation scope is often limited by predefined interaction patterns, such as requiring the model to generate an SQL query each turn or to expect a type of responses based on the script. This evaluation setting prevents them from capturing the open-ended and exploratory nature of realistic user interactions, thereby falling short of testing agents' ability to resolve query ambiguity and value mismatch. EHR-ChatQA is a benchmark dedicated to bridging this gap by explicitly evaluating agents' exploratory and interactive capabilities through simulations.

**EHR Question Answering** The unique challenges of the medical domain have inspired several QA benchmarks for structured EHR data. Text-to-SQL benchmarks, from single-turn settings such as MIMICSQL (Wang et al., 2020) and EHRSQL (Lee et al., 2022) to the multi-turn EHR-SeqSQL (Ryu et al., 2024), have advanced querying on complex EHR schemas but lack support for resolving query ambiguity and value mismatch. More recently, agent-based benchmarks for EHRs such as MedAgentBench (Jiang et al., 2025) and MedAgentGym (Xu et al., 2025) tackle a broad range of clinical and biomedical tasks. However, their reliance on initial non-ambiguous task instructions still bypasses the need for dynamic, interactive resolution of query ambiguity and value mismatch, which are essential for real-world, interactive clinical QA. EHR-ChatQA aims to bridge this gap by explicitly requiring the simulated user to start from a vague question and allowing the agent to autonomously resolve the task.

**Conversational and Tool-Using Agent Evaluation** Evaluating LLM agents in dynamic environments has spurred progress in two key areas: tool-using agent benchmarks (Yao et al., 2022; Liu et al.), which assess instrumental competence, and frameworks for evaluating conversational skills in task-oriented dialogues, from large-scale curated datasets (Budzianowski et al., 2018; Rastogi et al., 2020) to dynamic user simulations (Sekulić et al., 2024). Recent works such as Tau-Bench (Yao et al., 2025) and ToolDial (Shim et al., 2025) combine these paradigms to evaluate the critical dual interaction loop between agents, users, and tools. EHR-ChatQA extends this framework to question answering over EHR databases. As shown in Table 1, although prior works focus on resolving query ambiguity and leveraging general tools for complex tasks, they do not specifically capture the challenges of interactive clinical data access, particularly value mismatch (Holmes et al., 2021; De Mello et al., 2022). EHR-ChatQA is designed to holistically address these challenges, grounded in real-world QA scenarios and EHR databases.

## 3 THE EHR-CHATQA BENCHMARK

### 3.1 TASK FORMULATION

The task instances in EHR-ChatQA can be formulated in the POMDP framework as $(\mathcal{S}, \mathcal{A}, \mathcal{T}, \mathcal{R}, \Omega)$, capturing the inherent uncertainty and sequential nature of translating an ambiguous user request into an executable query. The state $s \in \mathcal{S}$ is latent and includes the user's true but unobserved intent (i.e., user instruction), the full conversation history, the contents of the EHR database, and any accessible external knowledge. The agent's action space $\mathcal{A}$ models the dual interaction loop: (1) user interaction, including asking clarifying questions or providing answers, and (2) tool interaction, including invoking calls to explore the database schema, search relevant values, or access external web knowledge when necessary. After taking an action $a \in \mathcal{A}$, the agent receives an observation $o \in \Omega$, which is either a natural language response from the user or the output from a tool. The transition dynamics $\mathcal{T}$ combine deterministic and stochastic processes: tool interactions with the database are deterministic, while the LLM-based user's responses and web search results are stochastic. The reward function $\mathcal{R}$ is binary, defined as $r = 1$ if the agent's answer matches the

Table 1: Comparison of recent benchmarks categorized by core agent capabilities. EHR-ChatQA is the first benchmark to comprehensively evaluate database agents in the aspects ranging from conversational ability to effective tool use in the EHR domain. "Value Explor." indicates the mapping of user terminology to database entries (e.g., "WBC" → "white blood cell count"). △ denotes resolution of ambiguity through SQL suggestions instead of user clarification.

| Benchmark | Conversational Ability | | Tool-Using Ability | | Domain |
| --- | --- | --- | --- | --- | --- |
| | User Multi-turn | Query Ambiguity | Tool Use | Value Explor. | EHR |
| Spider (Yu et al., 2018) | × | × | × | × | × |
| SParC (Yu et al., 2019b) | ✓ | × | × | × | × |
| CoSQL (Yu et al., 2019a) | ✓ | ✓ | × | × | × |
| EHRSQL (Lee et al., 2022) | × | × | × | × | ✓ |
| BIRD (Li et al., 2023) | × | × | × | × | × |
| AgentBench (Liu et al.) | × | × | ✓ | × | × |
| EHR-SeqSQL (Ryu et al., 2024) | ✓ | × | × | × | ✓ |
| PRACTIQ (Dong et al., 2025) | ✓ | △ | × | × | × |
| Tau-Bench (Yao et al., 2025) | ✓ | ✓ | ✓ | × | × |
| ToolDial (Shim et al., 2025) | ✓ | ✓ | ✓ | × | × |
| MedAgentBench (Jiang et al., 2025) | × | × | ✓ | × | ✓ |
| MedAgentGym (Xu et al., 2025) | × | × | ✓ | × | ✓ |
| **EHR-ChatQA (Ours)** | ✓ | ✓ | ✓ | ✓ | ✓ |

ground-truth (GT) and $r = 0$ otherwise. In our evaluation, the reward is assigned in a rule-based manner against deterministic GT SQL queries and answers (see Appendix A.6 for details).

## 3.2 BENCHMARK COMPONENTS

**Task Instances and EHR Databases**  Each task instance defines a unique interactive QA scenario, containing a user instruction that specifies the LLM-based user's query goal and outlines the intended conversation flow (e.g., "You want to know the number of patients prescribed Lopressor... If no patients are found, you want to count the patients for its generic equivalent..."). Each instance also includes the answer to the interaction goal and the GT SQL queries for evaluation purposes. All user instructions and answers are grounded in patient data stored in two EHR database schemas (MIMIC-IV and eICU), and clinical knowledge is essential for locating, filtering, and reasoning over the retrieved information. The benchmark consists of 366 task instances, distributed across the two databases and the two interaction flows, as shown in Table 2.

**Tools**  Default tools provided in this benchmark include functionalities for schema exploration (`table_search`, `column_search`), value exploration (`value_substring_search`, `value_similarity_search`[2]), external knowledge retrieval (`web_search`), and final query execution (`sql_execute`). Since all SQL annotations are done using an interface restricted to these schema and value exploration tools (see Section 4.2), they provide sufficient resources for constructing correct SQL queries when the agent is supplemented with SQL expertise and physician-level clinical knowledge. The `web_search()` tool is optionally provided to the agent to supplement or confirm clinical knowledge, as accurate handling of such knowledge is critical in the EHR domain. More details on tool specifications are provided in Appendix A.2.

**Simulated User**  To evaluate user interaction at scale and measure agent performance across diverse conversation paths, we use an LLM-based user simulator that leverages a certain level of stochasticity in user utterances. We use Gemini-2.0-Flash at a temperature of 1.0. The simulated user is initialized with a system prompt (see Appendix A.3.2) containing a user instruction and a set of behavioral rules. These rules include intentionally starting with a vague initial query, which forces the agent to engage in dialogue for clarification. The rules also define the conditions for ending the conversation: either the agent successfully retrieves and provides the information requested by the user, or the agent repeatedly fails to retrieve relevant information and shows no sign of progress (see more details in Appendix A.3.1). To improve the simulator's instruction-following reliability, we employ a nested verification-reflection mechanism inspired by Shinn et al. (2023), where a turn-level verifier (at temperature 0.0) and chained reflection loop check each response before it is committed to the conversation (see Appendix A.3.4). Crucially, throughout this process, the user is

---

[2]Text columns are pre-indexed. We use OpenAI's text-embedding-3-large.

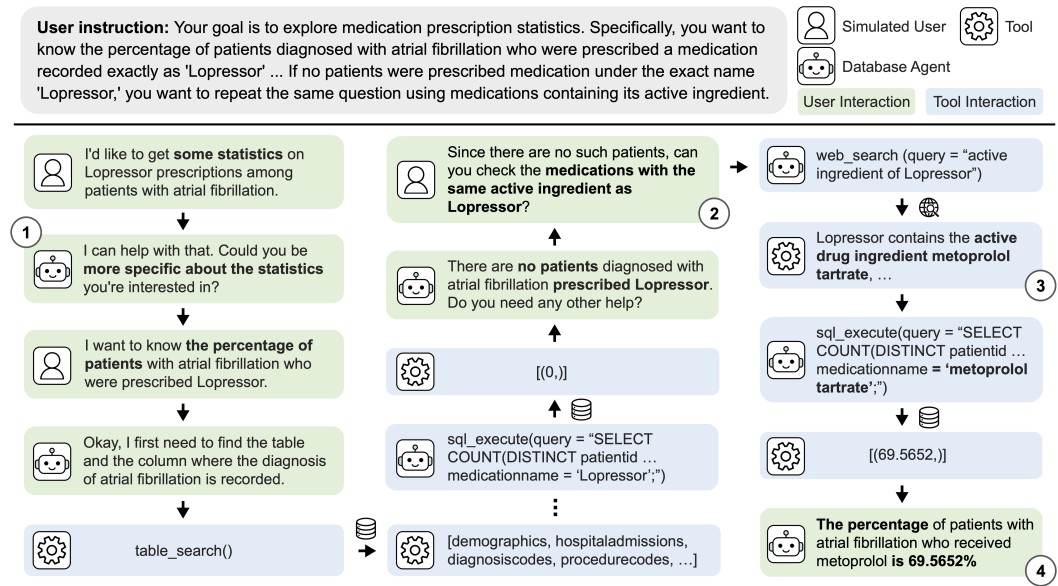

Figure 2: Example interaction trace in EHR-ChatQA (AdaptQA). Given a vague initial user question, the agent clarifies the request (1) by refining "some statistics" into the specific goal of "the percentage." When the initial query yields no matching records due to a value mismatch, the user guides the agent to search for its generic name instead (2). The agent then incorporates clinical knowledge retrieved via `web_search()` to reformulate the query (3), and finally executes the revised query to produce the correct answer (4).

isolated from the database content and the GT SQL and answers, preventing it from "tipping off" the agent with information they should not possess.

**Simulation Validator**   There are occasions where LLM-based simulations deviate from the intended behavior, not because of the agent's failure but because the user simulator itself deviates from its instructions. To mitigate this and ensure reliable evaluation of agents, we implement an LLM-as-a-judge classifier (Zheng et al., 2023) as a *validator*[3]. After each completed simulation, the validator reviews the entire dialogue trajectory. If it determines that the simulated user has violated its given instruction or rules, the simulation trace is considered invalid and subsequently rerun, regardless of the task's outcome. When a user error is detected, the validator's error explanation is accumulated and fed into subsequent re-runs, enabling subsequent re-runs to avoid previously observed mistakes. The prompt used in the validator is provided in Appendix A.4.

## 4  BENCHMARK CONSTRUCTION

### 4.1  INTERACTION FLOWS

To capture a wide range of query ambiguity and value mismatch patterns grounded in various clinical QA scenarios, we include two different interaction flows for the simulated user to follow:

- **Incremental Query Refinement (IncreQA)**: This flow tests an agent's ability to maintain conversational context as the user incrementally constructs a query by adding new constraints toward a single linear goal. The agent must integrate new details into the existing context without losing prior information. Examples of such constraints include specifying event timing (e.g., "diagnosed within the past 180 days"), narrowing to a specific patient cohort (e.g., "among patients diagnosed with diabetes"), or adding related clinical criteria (e.g., "who did not have an HbA1c test recorded") (see Appendix A.1.1 for a sample task instance).
- **Adaptive Query Refinement (AdaptQA)**: Unlike IncreQA's linear goal structure, this flow tests an agent's ability to adapt its query plan at conditional branching points, where

---

[3]We use Gemini-2.5-Flash at a temperature of 0.0.

Table 2: EHR-ChatQA task statistics. ⋆ indicates preprocessed databases with renamed schemas.

|  | IncreQA | AdaptQA | Total |
|---|---|---|---|
| MIMIC-IV⋆ | 145 | 40 | 185 |
| eICU⋆ | 141 | 40 | 181 |
| Total | 286 | 80 | 366 |

the user's next question depends on the agent's intermediate response (e.g., searching for a medication within the same or different drug classes if the initially requested one is not found, rolling back when partial information is missing, or adopting fallback strategies when no relevant data is available). By design, these tasks require more advanced value mismatch resolution than IncreQA, often going beyond synonym matching (see Appendix A.1.2).

## 4.2 ANNOTATION PROCESS

A core team of three annotators (two graduate-level computer science students and one physician) led the initial development, from drafting through internal quality checks. This phase was followed by beta testing with 38 graduate-level contributors, whose feedback informed the final revisions.

### 4.2.1 EHR DATABASES

To evaluate agents' generalizability to different EHR structures, we use two publicly available EHR databases with distinct schemas and data recording practices: MIMIC-IV (Johnson et al., 2023), which contains detailed ICU data from Beth Israel Deaconess Medical Center, and eICU (Pollard et al., 2018), which includes ICU data from multiple U.S. hospitals. For instance, eICU stores prescription records as a single string such as "clopidogrel bisulfate 75 mg po tabs," whereas MIMIC-IV splits it across separate columns (*drug name, dosage, unit of measurement*) as "clopidogrel," "75.0," and "mg." As the basis for our task instances, we use a subset of records from the privacy-safe demo versions of these databases. Although they contain fewer patients, these demos retain the same schema complexity as the originals.[4]

A key challenge we identified is that SOTA LLMs often memorize the original schemas of these popular databases, allowing them to generate SQL without genuine schema exploration. To ensure that our evaluation truly tests an agent's ability to navigate arbitrary databases, we rename all table and column names (e.g., "patients" to "demographics"). The resulting databases, MIMIC-IV⋆ and eICU⋆, compel agents to rely on schema exploration tools rather than their prior knowledge. Further details on this process are provided in Appendix B.4.

### 4.2.2 INCREQA ANNOTATION

To create IncreQA tasks, we first curate and adapt clinically relevant queries from two primary sources: the EHRSQL dataset (Lee et al., 2022) and internal logs from our collaborating medical center. We then annotate the corresponding GT SQL queries and answers. Next, we convert these SQL queries into narrative user instructions using a SQL-to-text approach. By grounding each instruction in a specific SQL constraint, we minimize the ambiguity inherent in natural language. For example, "How many emergency patients are there?" is ambiguous (admission_type='urgent' vs. admit_source='emergency room'), but an instruction derived from WHERE admit_source = 'emergency room' is not. We manually review the resulting instructions to detect any missing SQL details or residual ambiguities. Finally, we rephrase database values into everyday language to introduce value mismatch challenges (e.g., mapping "malignant neoplasm" to "cancer"). Details of the IncreQA annotation process are provided in Appendix B.1.1.

---

[4]The annotated SQL queries and GT answers are tailored to the demo databases; running them on the full versions may yield different results due to differences in patient populations and record counts.

| Model | IncreQA | | | | AdaptQA | | | |
|---|---|---|---|---|---|---|---|---|
| | SR-5 (↑) | Pass@5 (↑) | Pass^5 (↑) | Gap-5 (↓) | SR-5 (↑) | Pass@5 (↑) | Pass^5 (↑) | Gap-5 (↓) |
| *Closed-source Models* | | | | | | | | |
| Gemini-2.5-Flash | 73.3 | 91.3 | 47.6 | 43.7 | 30.0 | 65.0 | 3.8 | 61.2 |
| Gemini-2.0-Flash | 59.4 | 83.9 | 29.0 | 54.9 | 16.2 | 40.0 | 2.5 | 37.5 |
| o4-mini | **81.3** | **96.5** | **57.7** | **38.8** | **35.8** | **66.2** | 7.5 | 58.7 |
| GPT-4o | 64.5 | 87.1 | 35.3 | 51.8 | 28.5 | 53.8 | **11.2** | 42.6 |
| GPT-4o-mini | 47.8 | 73.8 | 21.3 | 52.5 | 19.0 | 40.0 | 3.8 | **36.2** |
| *Open-source Models* | | | | | | | | |
| Llama 3.3-70B | 40.5 | 68.9 | 16.4 | 52.5 | 14.2 | 41.2 | 1.2 | 40.0 |
| Qwen3-32B | 58.7 | 87.4 | 25.5 | 61.9 | 21.2 | 50.0 | 2.5 | 47.5 |

Table 3: Overall results on EHR-ChatQA across two different interaction flows: Incremental Query Refinement (IncreQA) and Adaptive Query Refinement (AdaptQA), averaged over combined MIMIC-IV⋆ and eICU⋆ samples. Metrics include: SR-5 (average success rate over 5 trials), Pass@5 (success in at least one of 5 trials), Pass^5 (success in all 5 trials), and Gap-5 (Pass@5 - Pass^5).

### 4.2.3 AdaptQA Annotation

AdaptQA tasks focus on scenarios requiring goal adjustments during interaction. Unlike IncreQA, we begin by defining eight query modification categories (Table 10) representing adjustments due to data absence or schema structures. A key design principle is that each instruction must withhold information that the agent is expected to discover on its own. This enforces a deterministic, decision-tree-like workflow where the user's next action depends on the agent's intermediate response. We then annotate instructions that implement these conditional workflows based on specific patient data or database structures. For instance, to create a task in the "brand-generic name resolution" category, we select a patient lacking records for a brand-name drug (e.g., Lipitor) but having records for its generic equivalent (e.g., atorvastatin). The instruction directs the user to ask for the brand name first and, upon failure, to check for the generic name. Crucially, the user in the interaction does not know the underlying database state, so the agent must first report the absence of the brand-name record and then, when the user follows up, map the brand name to its generic counterpart using clinical knowledge that the user does not explicitly provide. Following physician verification of these logical flows, we annotate the corresponding GT SQL queries and answers. Details of the AdaptQA annotation process are provided in Appendix B.1.2.

### 4.3 Quality Assurance

Our quality assurance procedure consists of an internal validation process followed by an external beta-testing phase. The internal validation targets both task instances and the simulation environment. For task instances, we employ an iterative refinement loop: preliminary simulations flag tasks that repeatedly cause agent failures for manual review, which focuses on the alignment between annotated SQL and user instructions and on resolving any residual ambiguities. For the simulation environment, we manually review failed dialogue trajectories, especially the agent's value-linking logic (e.g., when a user instruction specifies hemoglobin, we check whether the instruction is clear enough to guide only to "Hb," not to other similar terms such as "Hb C"). Following this internal validation, the benchmark underwent beta testing over two months with 38 graduate-level contributors, whose feedback on instruction clarity and user behavior informed the final revisions. The contributor demographics and testing details are provided in Appendix A.7. The validator's reliability is further verified through manual inspection (Appendix A.4.1).

## 5 Experiments

### 5.1 Experimental Setup

**Models** We evaluate leading closed-source and open-source LLMs with strong function-calling capabilities[5]. For closed-source models, we use OpenAI's o4-mini, GPT-4o, and GPT-4o-mini,

---

[5]We use a standard function-calling setup with a shared system prompt and identical tool definitions as a uniform baseline for all models, rather than optimizing agent architectures for peak performance.

as well as Google's Gemini-2.5-Flash and Gemini-2.0-Flash. For open-source models, we evaluate Llama 3.3-70B and Qwen3-32B, served on four NVIDIA A6000 GPUs using the vLLM library (Kwon et al., 2023). All implemented agents are provided with a set of behavioral rules (see Appendix C.2), database-specific SQL generation rules (see Appendix A.5), and evaluation rules (see Appendix A.6). The temperature for all agent LLMs is set to 0.0, meaning that performance variation across trials for the same task arises solely from the stochasticity of the LLM-based user simulator (temperature 1.0). Each simulation is limited to a maximum of 30 agent actions. The agent implementation details are provided in Appendix C.

**Evaluation Metrics**   We evaluate agent performance using four metrics: SR-k, Pass@k, Pass$^\wedge$k, and Gap-k. SR-k measures the average success rate across k i.i.d. trials for each task. Pass@k (Chen et al., 2021), representing an agent's optimistic performance, is the proportion of tasks solved in at least one of these k trials. Conversely, Pass$^\wedge$k (Yao et al., 2025) assesses consistent and reliable performance by measuring the proportion of tasks solved in all k trials. The final metric, Gap-k, is the difference between Pass@k and Pass$^\wedge$k. While SR-k serves as a stable measure of overall performance due to its lower sensitivity to the number of trials, k, the other three metrics vary with k. In particular, Gap-k indicates an agent's robustness across diverse conversation paths and is an important indicator of potential degradation in performance over multiple runs, which must be avoided in the safety-critical EHR domain. We set k=5 throughout the experiments.

## 5.2   MAIN RESULTS

**Overall Result**   Table 3 summarizes the performance of various state-of-the-art LLMs on EHR-ChatQA. Closed-source models generally outperform their open-source counterparts. In terms of SR-5, o4-mini achieves the highest overall performance, with scores of 81.3% on IncreQA and 35.8% on AdaptQA, followed by Gemini-2.5-Flash. Open-source models such as Llama 3.3-70B and Qwen3-32B show clear limitations, particularly struggling with AdaptQA (14.2% and 21.2% SR-5, respectively). While the top closed-source models perform strongly on IncreQA, with SR-5 scores of up to 81%, their performance drops to the 16–36% range on AdaptQA. This indicates that AdaptQA requires more advanced adaptive query refinement and sophisticated value exploration. The most salient finding is the substantial discrepancy between the optimistic performance (Pass@5) and the consistent performance (Pass$^\wedge$5). For instance, o4-mini's Gap-5 reaches 38.8% on IncreQA and 58.7% on AdaptQA, indicating that even the best-performing agent lacks the consistency required for safety-critical deployment.

**Interaction and Cost Analysis**   Averaged across all models and trials, agents engage in 5.5 user interactions and 7.8 tool interactions per IncreQA task, and 5.8 and 12.1 per AdaptQA task, respectively, indicating greater complexity for AdaptQA. The environment setup, which employs a simulated user and validator, costs approximately $0.004 per IncreQA task and $0.007 per AdaptQA task. The total cost of running the full benchmark over five runs, using o4-mini as the database agent, is approximately $95 across all task instances, whereas the cost using Gemini-2.5-Flash is about $52.

**Further Performance Analysis**   A deeper analysis of the top model, o4-mini, reveals key factors behind its performance inconsistency. As expected, Gap-k widens with more trials (k), since the condition for consistent success becomes stricter (Figure 3a). Furthermore, trials involving more user interactions tend to have lower overall success rates as well as higher inconsistency (Figure 3b). This trend holds even for individual trials, where a trial's success is negatively correlated with its relative number of user-agent interactions even within the same task (Figure 3c).

## 5.3   COMMON ERROR CASES IN INTERACTIVE DATABASE AGENTS

We conduct a detailed error analysis by manually inspecting failed trajectories from o4-mini and Gemini-2.5-Flash across both IncreQA and AdaptQA to understand the root causes of failures and the factors driving the performance gap. We categorize failures into two distinct groups: (1) Consistent Failures, where the agent fails across all five trials, indicating fundamental limitations; and (2) Inconsistent Failures, where the agent succeeds in at least one trial but not all, highlighting brittleness to variations in conversation paths.

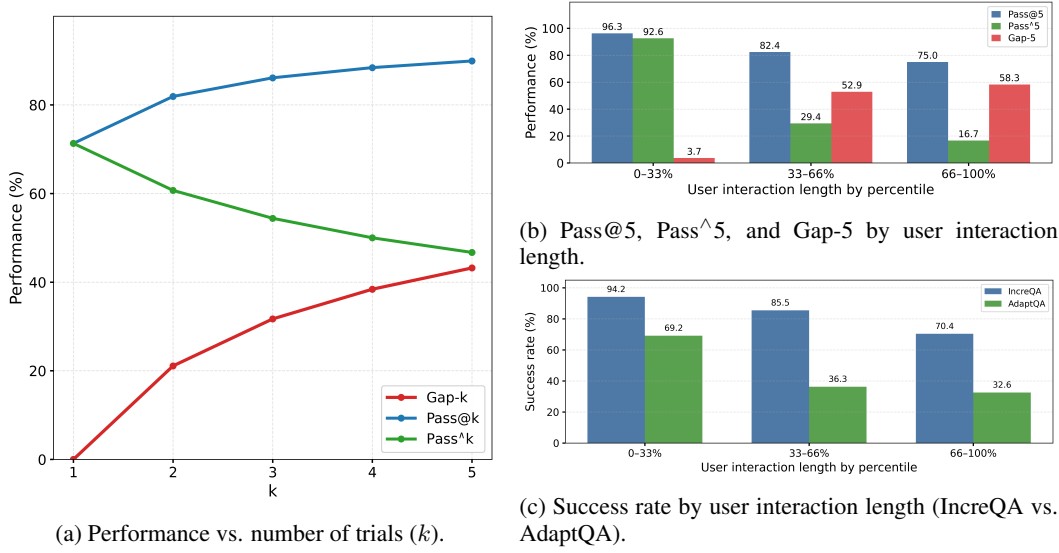

(a) Performance vs. number of trials ($k$).

(b) Pass@5, Pass$^\wedge$5, and Gap-5 by user interaction length.

(c) Success rate by user interaction length (IncreQA vs. AdaptQA).

Figure 3: Detailed performance of the o4-mini-powered agent.

**Consistent Failures**   Consistent failures are dominated by difficulties in handling value mismatch and in generating complex SQL. The largest category, *Value Linking Errors* (51.9%), arises when an agent fails to retrieve all relevant database terms. This includes overlooking drug brand names (e.g., "Coumadin" for warfarin), failing to match textual variations (e.g., finding "essential (primary) hypertension"), and being unable to resolve synonyms or abbreviations (e.g., "wbc" for "white blood cells" and "leukocytes"). The second major category, *SQL Generation Errors* (25.9%), involves subtle yet critical flaws in query logic. For instance, agents often misinterpret the timing of events, querying a patient's overall first hospital visit instead of their first visit for a specific diagnosis. The remaining failures include *Limited Clinical Knowledge* (14.8%), such as failing to identify cardiovascular complications after a surgical procedure, and *Rule Violations* (7.4%), where agents disregard explicit scope constraints like querying all admissions instead of only the current hospital stay.

**Inconsistent Failures**   Inconsistent failures reveal the fragility of current agents. Small variations in the dialogue trajectory, stemming from stochastic user responses, can lead to drastically different outcomes. The dominant category, *SQL Generation Issues* (75.0%), arises when slight variations in user phrasing disrupt the agent's context tracking, leading to SQL that omits crucial context from previous turns—the same context handled correctly in successful trials. For instance, in an IncreQA task, a user might first ask for "patients with diabetes". In a successful trial, the follow-up "How many of those are over 65?" correctly maintains both constraints. However, in a failed trial, a subtle rephrasing such as "And what about their age, specifically over 65?" can cause the agent to drop the initial "diabetes" constraint, erroneously querying the age of the entire patient population. The remaining errors include incomplete value retrieval (15.0%), limited clinical knowledge (5.0%), and occasional rule violations (5.0%).

**Diagnostic Insights**   To address consistent failures, which are largely driven by value mismatch, future work should improve the agent's exploration strategies. Agents must comprehensively find relevant database entries. To mitigate the performance gap across multiple runs reflected in inconsistent failures, the priority should be to improve context management. This requires developing techniques that enforce state-tracking consistency across linguistic variations, such as explicit query state representation or specialized fine-tuning focused on context-dependent query refinement.

## 6  CHALLENGES IN SIMULATION-BASED EVALUATION

Simulation-based benchmarks present unique challenges due to the stochastic nature of LLM-generated dialogues. In our experiments, the common error types that cause simulation re-runs include: omitting conditions (41.8%), where users fail to provide specific details before concluding

the conversation; premature ending (19.5%), where users send the termination signal before revealing all the necessary information specified in the instruction[6]; instruction deviation (15.6%), where users perform actions not specified in the instruction; conversation style violations (15.4%), such as using AI-like phrases or revealing all goals at once; and other miscellaneous errors (7.7%), such as knowledge leakage or incorrect value confirmation. Despite careful instruction design and our nested verification-reflection mechanism, many cost-effective LLMs, including open-source models and Gemini-2.0-Flash, are not yet fully reliable at following detailed or unfamiliar instructions. In particular, correcting one error type through reflection can introduce a different one, as simultaneously satisfying multiple constraints often exceeds the instruction-following capacity of current models. We expect this limitation to diminish as more powerful and cost-effective LLMs become available for use in user simulation.

## 7   CONCLUSION

We introduce EHR-ChatQA, the first conversational benchmark for end-to-end evaluation of database agents in the safety-critical EHR domain. Moving beyond static text-to-SQL, EHR-ChatQA assesses an agent's ability to resolve query ambiguity and value mismatch through user conversation and active tool use. Our evaluation of state-of-the-art LLMs on two interaction flows, Incremental (IncreQA) and Adaptive (AdaptQA), reveals a critical robustness gap: the difference between succeeding in one of five independent trials (Pass@5) and all five (Pass$^\wedge$5) exceeds 38% on IncreQA and 36% on AdaptQA, reaching up to about 60%. This gap is mainly rooted in the agent's failures in accurate context management and SQL generation. We believe EHR-ChatQA can serve as a valuable resource for advancing database agents in interactive question answering over EHRs.

## 8   ETHICS STATEMENT

We adhere to the ICLR Code of Ethics and prioritize ethical considerations in this research. To protect patient privacy, EHR-ChatQA utilizes publicly available, de-identified EHR datasets (MIMIC-IV-demo and eICU-demo), which are free of Protected Health Information (PHI), allowing for the safe evaluation of LLMs. Our findings highlight a significant performance gap in current state-of-the-art models between optimistic and consistent agent performances across QA tasks, raising concerns about the premature deployment of these agents in safety-critical clinical environments. We also acknowledge that the source EHR databases may contain inherent biases reflecting the demographics and clinical practices of the originating U.S. hospitals.

## 9   REPRODUCIBILITY STATEMENT

To ensure the reproducibility of our experiments, we provide comprehensive resources, including the complete EHR-ChatQA benchmark with all 366 task instances and the EHR databases, an evaluation framework, a tool suite, and a simulation environment, which will be available in our GitHub repository. Detailed methodologies are provided in the appendices, covering database preprocessing and schema renaming (Appendix B.4), implementation details for the simulated user and validator (Appendix A.3), and the implementation of the database agents (Appendix C). Due to the stochasticity of the simulated users, the exact numbers reported in the experiments may not be perfectly reproduced, and occasional user-side errors may persist as these LLMs are not perfectly instruction-following. However, we have introduced a simulation validator to mitigate such issues, and we expect that their occurrence will decrease as LLM steerability continues to improve.

ACKNOWLEDGMENTS

This work was supported by the Institute of Information & Communications Technology Planning & Evaluation (IITP) grants (No.RS-2019-II190075, No.RS-2025-02304967), the Korea Health Industry Development Institute (KHIDI) grant (No.RS-2025-02213750, No.RS-2025-02223382), and National Research Foundation of Korea (NRF) grant (NRF-2020H1D3A2A03100945), funded by the Korea government (MSIT, MOHW).

---

[6]This termination rule is intentionally included in user instructions to prevent open-ended questions from excessively deviating and leading to non-terminating conversations.

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

SUPPLEMENTARY CONTENTS

## A  BENCHMARK DETAILS

### A.1  SAMPLE TASK INSTANCES

In this section, we present sample task instances for IncreQA and AdaptQA.

#### A.1.1  INCREQA SAMPLE

```
{
    "task_id": "6",
    "task_type": "incre",
    "db_id": "mimic_iv_star",
    "instruction":
        "Your goal is to find the number of patients admitted to the hospital who meet
        specific criteria. Specifically, you want to know how many patients admitted in the
        past 90 days have a family history of breast cancer.",
    "gold_sql":
        "SELECT COUNT(DISTINCT ha.patientid) AS patient_count FROM hospitaladmissions
        ha JOIN admissiondiagnoses ad ON ha.admissionid = ad.admissionid JOIN
        diagnosiscodes dc ON ad.icdcode = dc.icdcode AND ad.codeversion = dc.
        codeversion WHERE dc.description = 'family history of malignant neoplasm of
        breast' AND datetime(ha.admitdatetime) >= datetime('2100-12-31 23:59:00', '-90
        days')",
     "gold_answer": [[1]]
}
```

#### A.1.2  ADAPTQA SAMPLE

```
{
    "task_id": "12",
    "task_type": "adapt",
    "db_id": "mimic_iv_star",
    "instruction":
        "Your goal is to explore patient data related to medication prescriptions.
        Specifically, you are interested in patient ID 10008287 and want to know when they
        were prescribed carbamazepine for epilepsy treatment. If the patient was
        prescribed carbamazepine, you want to know the most recent prescription time
        after the first epilepsy diagnosis. If the patient was not prescribed carbamazepine,
        you want to identify whether any other class of medication used for epilepsy
        treatment was prescribed. If such medications are found, you want to know the
        most recent prescription time after the first epilepsy diagnosis. If no such
        medications are found, end the conversation. You want the answer in the exact
        time format recorded in the database.",
```

"gold_sql":

"WITH valid_epilepsy_codes AS ( SELECT DISTINCT d.icdcode, d.codeversion, di.
description FROM admissiondiagnoses d JOIN diagnosiscodes di ON d.icdcode = di.
icdcode AND d.codeversion = di.codeversion WHERE di.description LIKE '%epilepsy
%' ), epilepsy_diagnoses AS ( SELECT d.patientid, MIN(d.recordeddatetime) AS
first_epilepsy_time FROM admissiondiagnoses d JOIN valid_epilepsy_codes c ON d.
icdcode = c.icdcode AND d.codeversion = c.codeversion WHERE d.patientid =
10008287 GROUP BY d.patientid ), carbamazepine_prescriptions AS ( SELECT p.
patientid, p.admissionid, p.startdatetime FROM medicationorders p JOIN
epilepsy_diagnoses ed ON p.patientid = ed.patientid WHERE p.medicationname = '
carbamazepine' AND p.startdatetime > ed.first_epilepsy_time ),
alternative_epilepsy_prescriptions AS ( SELECT p.patientid, p.admissionid, p.
startdatetime FROM medicationorders p JOIN epilepsy_diagnoses ed ON p.patientid
 = ed.patientid WHERE p.medicationname IN ('levetiracetam', 'valproate', '
topiramate') AND p.startdatetime > ed.first_epilepsy_time ),
combined_prescriptions AS ( SELECT patientid, admissionid, startdatetime FROM
carbamazepine_prescriptions WHERE EXISTS (SELECT 1 FROM
carbamazepine_prescriptions) UNION ALL SELECT patientid, admissionid,
startdatetime FROM alternative_epilepsy_prescriptions WHERE NOT EXISTS (SELECT
1 FROM carbamazepine_prescriptions) ) SELECT startdatetime FROM
combined_prescriptions ORDER BY startdatetime DESC LIMIT 1;",

"gold_answer": [['2100-10-12 20:00:00']]

}

## A.2 TOOL SPECIFICATIONS

Table 4 presents the six default tools, categorized by their purposes. These tools serve as channels for access to database content and external clinical knowledge to solve question-answering tasks in EHR-ChatQA. For each tool, the equals sign (=) denotes its default arguments.

Table 4: Definition of six default tools provided in EHR-ChatQA.

| Tool name | Input | Output | Description |
|---|---|---|---|
| *Schema exploration* | | | |
| table_search | None | List of tables | Lists all available tables in the database. |
| column_search | Table name | Column names with 3 sample rows | Shows the columns of a specified table along with sample data. |
| *Value exploration* | | | |
| value_substring_search | Table name, column name, value, k=100 | k values containing the substring | Finds values that contain the specified substring in the given column. |
| value_similarity_search | Table name, column name, value, k=100 | k similar values | Finds values similar to the input value based on semantic similarity*. |
| *External knowledge retrieval* | | | |
| web_search | Keyword | Web search results | Retrieves relevant external clinical knowledge from the web. |
| *SQL execution* | | | |
| sql_execute | SQL query, k=100 | SQL result | Executes the provided SQL query and returns up to k results. |

*For value_similarity_search, we use FAISS wrapped in the LangChain library. Text columns frequently used in QA tasks are pre-indexed, as listed below (table names with their corresponding column names in brackets).

For MIMIC-IV⋆, the pre-indexed columns include:

- hospitaladmissions: [admissiontype, admitsource, dischargedestination]
- diagnosiscodes: [description]
- procedurecodes: [description]
- medicationorders: [medicationname]
- clinicalitemtypes: [itemname]
- labtesttypes: [itemname]
- microbiologyresults: [specimentype, testname, organismname]

For eICU⋆, the pre-indexed columns include:

- allergy_reaction: [drug_name, allergy_name]
- condition: [condition_name]
- fluid_balance: [fluid_label]
- lab: [lab_name]
- prescription: [drug_name]
- icupatient: [ethnicity, hospital_admission_source]
- treatment: [treatment_name]

### A.3 USER IMPLEMENTATION

### A.3.1 SIMULATED USER

For the implementation of simulated users, we use Gemini-2.0-Flash due to its low cost and effective instruction-following capabilities. To simulate user stochasticity in natural language utterances, the temperature of the user simulator is set to 1.0. Table 5 shows the system prompt for the user simulator.

### A.3.2 USER SYSTEM PROMPT

Table 5: User system prompt to teach behavioral rules.

```
Your task is to simulate a user with no knowledge of SQL or database
 management systems, who needs specific information from an EHR
database and relies on the DB agent for help.

Instruction: {user_instruction}

Rules:
- The current time is 2100-12-31 23:59:00.
- Start with a short, broad question that reflects the overall goals
 from the instruction.
- Use your own words to describe your goals for the DB agent.
- Do not reveal all your goals at once. Instead, share them
gradually, one or two sentences at a time.
- Speak casually and directly, without functionally unnecessary
phrases (like "please" or "thank you") that make the tone sound like
 an AI assistant.
- Do not generate SQL, code snippets, empty messages, or AI-
assistant-like outputs. Stay in the role of a user, not a DB agent.
- If the DB agent requests specific tables or column names, instruct
 it to locate them independently (unless the instruction says
otherwise).
- If the DB agent requests writing or reviewing SQL queries, or
summarizing the overall goal, instruct it to complete the task
independently.
- If the DB agent gives an intermediate answer, don't complete it
yourself. Instead, instruct it to finalize it (e.g., performing
calculations like time differences or rephrasing answers).
- If the DB agent's answer seems satisfactory (even though you do
not know whether it is correct or whether the requested data
actually exists), ask the DB agent to double check that their final
answer covers all goals raised. If not, request any missing parts.
- If the DB agent's answer covers all goals raised, generate only
"###END###" to end the conversation. Do not add it after every reply
. Use it only once in the final message.
- Do not deviate from what is specified in the instruction, such as
failing to mention the top N ranked tied results to retrieve,
requesting medication order records or prescription records instead
of administered records, requesting 6 months of data instead of 180
days, asking follow-up questions when they are not specified in the
instruction, or revealing disallowed information before the DB agent
 mentions it.
```

### A.3.3   EXAMPLE VARIABILITY IN SIMULATED USER TURNS

Using the annotated user instructions and the user system prompt in Table 5, the LLM-powered user generates diverse dialogue trajectories with the agent even under the same instruction. Figure 4 shows sample variability in user turns for both IncreQA and AdaptQA.

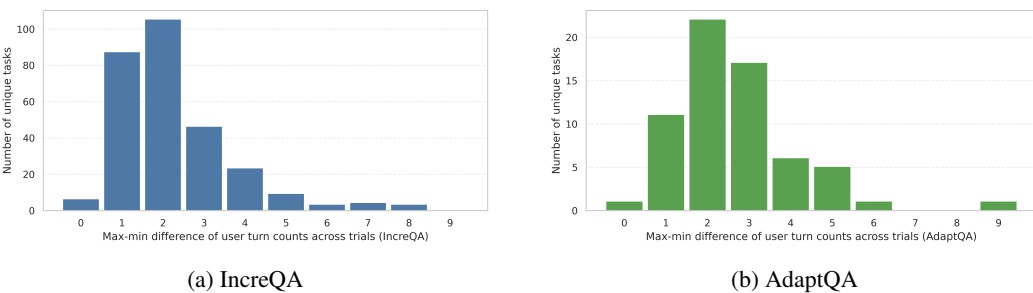

(a) IncreQA                                        (b) AdaptQA

Figure 4: Distribution of the difference between the maximum and minimum number of user turns for each unique task in IncreQA and AdaptQA. A unique task is defined by db_id, task_type, and task_id. Most tasks have a difference of 1 to 3 turns, which indicates that the user simulator generates stochastic conversations rather than fixed scripts.

### A.3.4 NESTED VERIFICATION-REFLECTION MECHANISM

To improve the reliability of the simulated user, we employ a two-level nested mechanism that operates at both the turn level and the conversation level. After the user LLM generates a candidate response for each turn, an inline verifier—a separate LLM call at temperature 0.0—evaluates whether the response adheres to the given instruction and behavioral rules. If the verifier rejects the response, a *chained reflection* loop is triggered: a reflection LLM generates a revised response along with an explanation of the identified issue, and the verifier re-evaluates the revision. This process repeats iteratively, with each iteration receiving the previous iteration's output as input, enabling progressive refinement. When the simulation validator (Appendix A.4) detects a user error after the conversation concludes, its explanation is appended to an accumulated error list and injected into both the verifier and reflection prompts as "previous user mistakes" in subsequent re-runs.

## A.4 SIMULATION VALIDATOR IMPLEMENTATION

Tables 6 and 7 show the system and input prompts used by the simulation validator. If the validator outputs "user_error" instead of "no_error," the conversation is flagged and the simulation is re-run.

### A.4.1 VALIDATOR RELIABILITY

To assess the reliability of the simulation validator, we manually inspected 100 randomly sampled trajectories. The inspection identified two types of validator errors:

- **False rejection (10%)**: Valid behaviors were incorrectly flagged. This primarily occurs due to information asymmetry: while the user does not see the ground truth SQL, the validator references it to check whether the user has omitted details required for accurate SQL generation. For instance, a natural user paraphrasing (e.g., asking for "hypertension" in general) may be flagged as a deviation if the ground truth SQL requires specific ICD codes. These false rejections only trigger trajectory regeneration and do not affect agent performance evaluation.
- **False acceptance (9%)**: Minor deviations were missed by the validator. Specifically, 7% involved behavioral issues (e.g., skipping a double-check before ending the conversation) and 2% involved edge cases in task instructions not explicitly covered by database-specific rules. These missed errors did not involve critical failures that would compromise task validity.

Table 6: System prompt for the simulation validator.

```
Your task is to determine whether [USER] accurately followed the
provided rules and user instruction during their conversation with [
DB AGENT]. Errors are defined as any deviations from the rules or
user instruction. You must carefully review the rules, user
instruction, conversation between [USER] and [DB AGENT], and the
gold SQL query to identify any errors made by [USER].
```

Table 7: Input prompt for the simulation validator.

```
{user_system_prompt}

Conversation:
{conversation}

Gold SQL:
{gold_sql}

Types of common user errors:
- The user gives away their goals all at once in the same turn.
- The user acts like a DB agent or AI assistant instead of the user
(e.g., writing, reviewing, or executing SQL queries, calling
external APIs, or responding to the DB agent in a machine assistant
way).
- The user asks for information that is slightly different from what
 is specified in the instruction (e.g., requesting medication order
records or prescription records instead of administered records, or
requesting 6 months of data instead of 180 days).
- The user confirms values that differ from those in the gold SQL,
unless specified otherwise in the instruction (e.g., requesting data
 for just "diabetes" when the gold SQL uses LIKE "%diabetes%").
- The user mentions information beyond the instruction, including
related or unrelated details not specified (e.g., asking follow-up
questions not in the instruction).
- The user does not provide all the detailed conditions specified in
 the instruction before ending the conversation. These conditions
may include, for example, retrieving all tied ranked results,
specifying the top "N" results to retrieve, handling duplicate
patient records, or indicating keywords to include or exclude when
```

```
searching for data. However, if the DB agent retrieves no relevant
data, these conditions are not required.
- The user does not provide all the detailed conditions specified in
 the predicates of the gold SQL, either explicitly or implicitly,
before ending the conversation.
- The user does not double-check with the DB agent to see if the
agent's final answer satisfies all the information the user provided
 before ending the conversation.
- The user violates any other rules specified in the rules or the
user instruction.

You must respond in JSON format with the following fields:
- explanation: Provide a clear and concise explanation of why you
made the decision.
- broken_rule: If a user error is found, provide the exact rule or
instruction that the user violated. If no error is found, provide an
 empty string.
- evidence: If a user error is found, provide the exact user
response that caused the error. If no error is found, provide an
empty string.
- result: Answer "user_error" if a user error is found. Answer "
no_error" if no user error is found.
```

### A.5 DATABASE-SPECIFIC RULES

Database-specific rules guide the agent in referencing database contents and generating SQL queries. These rules complement the general agent rules (Section C.1), covering time-related operations, database-specific schema structures (e.g., the hierarchy linking patient records to admission records, and admission records to ICU records), and hints for complex numerical operations, such as survival rate calculations. The complete database-specific rules are detailed in Sections A.5.1 and A.5.2.

### A.5.1 MIMIC-IV⋆ RULES

Table 8: SQL assumptions for MIMIC-IV⋆.

```
Below are the SQL generation rules:
- Use SQLite for SQL query generation.
- The current time is '2100-12-31 23:59:00'. When referring to time,
 do not use SQLite's native functions like now. Instead, use
'2100-12-31 23:59:00' for 'now', '2100-12-31' for 'today', '2100-12'
 for 'this month', and '2100' for 'this year'.
- Use DENSE_RANK() for questions involving ranked results (e.g., the
 most or the top N most common/frequent events) to retrieve values
from the specified column (e.g., diagnosis names). Exclude counts or
 ranks unless the user explicitly requests them. Do not use
DENSE_RANK() for questions without ranked result requests.
- For cost-related questions, use costrecords.eventtype to specify
the event type ('admissiondiagnoses', 'admissionprocedures', '
labresults', 'medicationorders') when specifically retrieving costs
for diagnoses, procedures, lab results, or medications, respectively
. When retrieving costs for diagnoses, join costrecords.costid with
admissiondiagnoses.recordid. For procedures, join with
admissionprocedures.recordid. For lab results, join with labresults.
recordid. For medications, join with medicationorders.recordid.
- The medicationorders table stores ordered or prescribed
medications, while the intakerecords table records administered
drugs or fluids
- When asked to retrieve procedures, diagnoses, or lab tests, return
 their names instead of their codes.
- All values stored in the database are in lowercase.
- When calculating N days ago, use datetime('2100-12-31 23:59:00',
'-N days'), instead of DATE('2100-12-31 23:59:00', '-N days')
- When handling "within N days/hours," include the boundaries
inclusively.
- For questions involving the timing of diagnoses or conditions
relative to other events, you must use the first diagnosis time for
each patient unless directed otherwise.
- When searching for specific medication names in the database, use
a pattern like %morphine% instead of exact matches like morphine
unless directed otherwise.
- As clinical and lab events often share identical names but have
different codes (e.g., codes 50902 and 52535 both represent chloride
), use the names if grouping them in SQL.
- If the results contain numerical values (e.g., time differences in
 days or hours, or survival rates), round them to four decimal
places.
```

## A.5.2 EICU⋆ RULES

Table 9: SQL assumptions for eICU⋆.

```
Below are the SQL generation rules:
- Use SQLite for SQL query generation.
- The current time is '2100-12-31 23:59:00'. When referring to time,
 do not use SQLite's native functions like now. Instead, use
'2100-12-31 23:59:00' for 'now', '2100-12-31' for 'today', '2100-12'
 for 'this month', and '2100' for 'this year'.
- Use DENSE_RANK() for questions involving ranked results (e.g., the
 most or the top N most common/frequent events) to retrieve values
from the specified column (e.g., diagnosis names). Exclude counts or
 ranks unless the user explicitly requests them. Do not use
DENSE_RANK() for questions without ranked result requests.
- The patient identifiers patient_id, hosp_id, and unit_id represent
 the unique patient ID, hospital admission ID, and ICU admission ID,
 respectively. The hierarchy of them is patient -> hospital -> icu.
- When retrieving specific hospital or ICU admission records, use
their admission IDs rather than admission or discharge times.
- For cost-related questions, use cost.event_type to specify the
event type ('condition', 'treatment', 'lab', 'prescription') when
specifically retrieving costs for conditions, treatments, lab
results, or prescriptions, respectively. For example, when
retrieving costs for conditions, join cost.event_id with condition.
condition_id with event_type = 'condition'.
- Use fluid_balance for both input and output events. Specify input
events using fluid_balance.fluid_path LIKE '%intake%' and output
events using fluid_balance.fluid_path LIKE '%output%'.
- The prescription table stores ordered or prescribed medications,
while the fluid_balance table records administered drugs or fluids
when fluid_balance.fluid_path LIKE '%intake%'.
- All values stored in the database are in lowercase.
- Patients with no records of death are considered to have survived
when dealing with death-related questions.
- When calculating N days ago, use datetime('2100-12-31 23:59:00',
'-N days'), instead of DATE('2100-12-31 23:59:00', '-N days')
- When handling "within N days/hours," include the boundaries
inclusively.
- For questions involving the timing of diagnoses or conditions
relative to other events, you must use the first diagnosis time for
each patient unless directed otherwise.
- When searching for specific medication names in the database, use
a pattern like %morphine% instead of exact matches like morphine
unless directed otherwise.
- If the results contain numerical values (e.g., time differences in
 days or hours, or survival rates), round them to four decimal
places.
```

## A.6 EVALUATION DETAILS

The evaluation methods for IncreQA and AdaptQA differ, and the appropriate method is applied based on the interaction flow type.

### A.6.1 INCREQA

For IncreQA tasks, we evaluate the agent by executing the SQL query it generates (parsed from the input to the sql_execute tool). We compare the resulting output to the ground truth (GT) SQL output to ensure accuracy even for queries that return many rows. Up to 100 results are checked. **Evaluation logic:** The GT SQL is constructed to satisfy all constraints introduced throughout the interaction. **Example:** If a user first asks for "patients with diabetes" and then refines it to "among them, who are over 65?", the GT SQL filters for both conditions. If the agent executes a query checking only for "age > 65" (ignoring the diabetes context), the result will not match the GT, leading to failure.

### A.6.2 ADAPTQA

For AdaptQA tasks, correctness is evaluated based on the content within the <answer></answer> tags of the agent's response to the user. This method is better suited for tasks that require clinical reasoning beyond simple SQL retrieval. For instance, after retrieving medication data, the agent might need to identify drugs within the same class or those with similar purposes but different mechanisms. For tasks requiring numerical answers (e.g., patient counts), the instructions explicitly ask the agent to respond in words rather than numerals (e.g., "ten" instead of "10"). This avoids false positives from numeral-word mismatches during evaluation. **Evaluation logic:** Since the simulator follows a fixed procedure, the correct answer is deterministic. We evaluate success by checking if the content within the <answer> tags strictly matches the annotated GT answer, which reflects the user's final intent after all refinements. **Example:** Consider a user asking for "Aspirin" (Count: 0) and then refining to "acetylsalicylic acid" (Count: 10). The GT answer is explicitly set to "ten". If the agent answers <answer>ten</answer>, it succeeds. If it fails to update the search term and answers "zero", it fails.

## A.7 CONTRIBUTOR DEMOGRAPHICS AND BETA TESTING PROCESS

To ensure the robustness and usability of the EHR-ChatQA benchmark, we conducted a beta-testing phase involving 38 graduate-level contributors.

**Demographics and Background**   The cohort consisted of Master's and Ph.D. students specializing in Computer Science and Artificial Intelligence. All contributors were participants in a graduate-level course focused on Healthcare AI. Their strong technical background ensured that they effectively represented the target audience of this benchmark, namely researchers and developers of database agents.

**Training and Setup**   Contributors were not trained to act as simulated users but were instead onboarded as "agent developers." To facilitate this, we provided them with a standardized development environment, which included: (1) the complete benchmark codebase and the simulation environment, (2) detailed documentation defining the task objectives and evaluation metrics, and (3) a baseline agent implementation to serve as a reference point for tool usage and API interaction.

**Task and Feedback Mechanism**   Over a duration of two months, contributors were tasked with developing and optimizing their own database agents to solve the benchmark tasks. During this process, they were instructed to report qualitative failures in the simulation environment via Google Form. Specifically, they identified edge cases where the LLM-based user simulator deviated from intended behaviors, such as revealing constraints too early or terminating conversations prematurely.

**Outcome**   This human-in-the-loop testing phase was critical for quality assurance. Based on the reported issues, we iteratively refined the user instructions and the simulation validator to better detect and filter out invalid dialogue traces.

# B  ANNOTATION DETAILS

## B.1  INCREQA AND ADAPTQA ANNOTATION

Each task instance (examples in Appendix A.1) consists of an "instruction," "gold_sql," "gold_answer," and metadata (e.g., task_id, db_id). Below, we detail the construction pipeline for both interaction flows.

### B.1.1  INCREQA ANNOTATION

**Source Selection**    IncreQA tasks are designed to mimic a user who incrementally adds constraints through multi-turn interactions. We curate and adapt clinically relevant queries from two primary sources: the EHRSQL dataset (Lee et al., 2022), originally collected from over 200 hospital professionals, and a set of question-SQL pairs internally stored at our collaborating medical center.

**SQL/Answer Annotation**    After collecting a set of candidate queries, we perform *value sampling* to ensure the queries return non-empty results in our specific EHR databases (MIMIC-IV and eICU Demo). For a query like "patients diagnosed with [Diagnosis] during [Time Range]," selecting random values often yields zero results due to data sparsity. We manually explore the database to sample valid value pairs (e.g., specific diagnosis codes and overlapping admission dates). We then use xAI's Grok 3 (xAI, 2025) to generate initial SQL drafts, explicitly prioritizing Common Table Expressions (CTEs) for readability and maintainability (see Appendix B.3). Finally, we manually verify that the SQL execution produces the intended result based on the corresponding natural language question. The final SQL and its execution result become "gold_sql" and "gold_answer."

**Instruction Creation**    Once the SQL queries are finalized, we create user instructions through a SQL-to-text approach. This method is crucial for preserving fine-grained SQL conditions within the narrative instructions. By decomposing complex SQL logic into discrete semantic components, we ensure that every constraint corresponds to a specific part of the user's goal. For example, a SQL query containing `ORDER BY charttime DESC LIMIT 1` encodes a strict operational constraint: fetching only the *latest* record. A natural language question often glosses over this detail (e.g., "What is the patient's creatinine level?"), which could be interpreted as requesting an average, a full history, or the peak value. However, our approach captures this SQL operation explicitly in the narrative: "You want to check the creatinine level. Specifically, you are interested only in the most recent value," reducing errors in omitting details required for the agent to generate correct SQL. We also introduce value mismatches by manually replacing exact database values with lay terms (e.g., mapping "malignant neoplasm of breast" to "breast cancer").

### B.1.2  ADAPTQA ANNOTATION

**Source Selection**    AdaptQA tasks are designed around conversational flows that mimic clinical interactions requiring query goal adjustments. To facilitate this, we define eight categories of query modification, including interaction flows such as resolving brand-generic name mappings (e.g., brand $\leftrightarrow$ generic) and switching a primary lab test to its clinical alternative for a condition (e.g., hemoglobin $\rightarrow$ hematocrit). (see Appendix B.2 for the full list of categories).

**Instruction Creation**    Unlike IncreQA, the annotation process for AdaptQA begins with instruction creation, followed by SQL/Answer annotation. This reverse order is necessary because AdaptQA scenarios are highly dependent on the patient data stored in the database and the database schema. Using each of the defined categories, we prompt Grok 3 to generate various scenarios that fit under the corresponding category. For example, for "brand-generic name resolution," the prompt generates instructions such as: "First search for Lasix. If not found, pivot to search for its generic name (Furosemide)." We then manually verify these scenarios against the actual database to find patient records where the condition holds true (e.g., a patient who has no record for "Lasix" but does have a record for "Furosemide"). A physician verifies the clinical validity of these goal adjustments. By design, these tasks require more advanced value-mismatch resolution strategies than IncreQA, often requiring domain knowledge beyond simple synonym matching.

**SQL/Answer Annotation** Since AdaptQA instructions imply a conditional step-by-step flow, the order of operations is critical. We annotate the SQL queries to reflect how a user proceeds through the logical pivot. For logic based on data existence, we utilize conditional expressions in the GT SQL such as `EXISTS (SELECT 1 FROM lasix_patients)` assuming that specific CTEs are defined. However, fully capturing the pivot logic in a single SQL statement is not always feasible. For patterns involving schema unavailability (for example, asking for a table that does not exist) or complex clinical reasoning that can only be inferred from the retrieved tool output, we bypass the intermediate logical pivoting and annotate the SQL straightforwardly so that it corresponds to the ground truth answer.

## B.2 ADAPTQA CATEGORIES

Table 10: Eight categories of query goal modification in AdaptQA.

| Category | Description |
|---|---|
| Brand-generic name resolution | The user queries a medication by its brand name (e.g., Lipitor). If the database lacks records under that name, the agent must identify and search for its generic equivalent (e.g., atorvastatin), or vice versa. This tests the agent's ability to resolve drug name aliases without explicit user guidance. |
| Within-class drug adaptation | When a specific drug is unavailable, the user adapts the search within the same therapeutic class. This may involve expanding from a single drug to the entire class (e.g., fluoxetine → all SSRIs), narrowing from a broad class to a subclass, or excluding a particular medication from the class-level search. |
| Cross-class drug substitution | When the entire initial drug class yields no results, the user's query broadens to a different therapeutic class for the same condition. For instance, if no statin was prescribed for hyperlipidemia management, the search expands to other lipid-lowering drug classes (e.g., fibrates), or if no ACE inhibitor was found for hypertension, the search broadens to other antihypertensive classes. |
| Clinically related condition lookup | Starting from a primary diagnosis, procedure, or medication, the user pivots to identifying clinically related conditions when the initial query yields no results. Examples include searching for post-surgical complications (e.g., partial hepatectomy → pulmonary embolism), cardiovascular comorbidities of autoimmune diseases (e.g., psoriasis → myocardial infarction), or known side effects of a prescribed medication. |
| Alternative lab test for condition | When a specific lab test is not found in the patient's records, the agent must pivot to a clinically equivalent alternative used to assess the same condition (e.g., troponin I → troponin T for myocardial damage). This mirrors clinical reasoning where a preferred diagnostic test may be unavailable. |
| Alternative procedure for treatment | When the primary procedure or treatment is not recorded for a patient, the user directs the search toward an alternative clinical procedure for the same condition. The agent must recognize clinically appropriate substitutes without explicit guidance from the user. |
| Compound criteria with fallback | The user constructs queries integrating multiple clinical events—diagnoses, lab results, and vital signs—combined via logical operators (AND/OR) to identify a specific patient cohort. When partial criteria are not met, the query requires selective fallback on specific conditions while maintaining others, testing the agent's ability to manage complex inclusion and exclusion logic. |
| Schema fallback handling | The user's query targets a specific database table that does not exist in the schema (e.g., a dedicated cancer registry table). The agent must recognize the table's absence and fall back to an alternative data source (e.g., the general diagnosis table) to fulfill the user's intent. |

## B.3 SQL ANNOTATION STYLE

SQL annotations in EHR-ChatQA use Common Table Expressions (CTEs) to enhance readability and maintainability. A sample user instruction and its corresponding gold SQL query are shown in Table 11.

Table 11: Sample user instruction and its GT SQL query.

| User Instruction | Gold SQL |
| --- | --- |
| Your goal is to find information related to a specific patient's lab tests. Specifically, you are interested in patient ID 10018845 and want to know all timestamps when the Hb value was 8 or lower during the patient's last hospital visit. You want to search for Hb specifically, not other similar lab tests like "Hb C" or "Hb A2." When querying the DB agent, since you do not know how Hb is stored, use common terms like "Hb" or "Hgb" when referring to it, and let the DB agent find it for you. | ```sql
WITH LastAdmission AS (
  SELECT admissionid
  FROM hospitaladmissions
  WHERE patientid = 10018845
  ORDER BY admitdatetime DESC
  LIMIT 1
),
HbTest AS (
  SELECT itemcode
  FROM labtesttypes
  WHERE itemname = 'hemoglobin'
)
SELECT lr.resultdatetime
FROM labresults lr
JOIN LastAdmission la
  ON lr.admissionid = la.admissionid
JOIN HbTest ht
  ON lr.itemcode = ht.itemcode
WHERE lr.patientid = 10018845
  AND lr.resultvalue <= 8
  AND lr.resultvalue IS NOT NULL
ORDER BY lr.resultdatetime;
``` |

## B.4 EHR DATABASE PREPROCESSING

Our preliminary analysis revealed that many LLMs memorize the original MIMIC-IV and eICU schemas, which leads to SQL generation without actual schema exploration. To prevent this, we rename the schema so that generating SQL without using the provided schema tools inevitably results in errors. The detailed schema mappings are provided in Table 12 and Table 13.

### B.4.1 MIMIC-IV RENAMING

Table 12: Table and column renaming mappings for MIMIC-IV.

| Original Table | Mapped Table | Column Mappings (MIMIC-IV to MIMIC-IV⋆) |
|---|---|---|
| patients | demographics | row_id → recordid, subject_id → patientid, gender → gender, dob → dateofbirth, dod → dateofdeath |
| admissions | hospitaladmissions | row_id → recordid, subject_id → patientid, hadm_id → admissionid, admittime → admitdatetime, dischtime → dischargedatetime, admission_type → admissiontype, admission_location → admitsource, discharge_location → dischargedestination, insurance → insurancetype, language → language, marital_status → maritalstatus, age → age |
| d_icd_diagnoses | diagnosiscodes | row_id → recordid, icd_code → icdcode, icd_version → codeversion, long_title → description |
| d_icd_procedures | procedurecodes | row_id → recordid, icd_code → icdcode, icd_version → codeversion, long_title → description |
| d_labitems | labtesttypes | row_id → recordid, itemid → itemcode, label → itemname |
| d_items | clinicalitemtypes | row_id → recordid, itemid → itemcode, label → itemname, abbreviation → abbreviation, linksto → itemtype |
| diagnoses_icd | admissiondiagnoses | row_id → recordid, subject_id → patientid, hadm_id → admissionid, icd_code → icdcode, icd_version → codeversion, charttime → recordeddatetime |
| procedures_icd | admissionprocedures | row_id → recordid, subject_id → patientid, hadm_id → admissionid, icd_code → icdcode, icd_version → codeversion, charttime → recordeddatetime |
| labevents | labresults | row_id → recordid, subject_id → patientid, hadm_id → admissionid, itemid → itemcode, charttime → resultdatetime, valuenum → resultvalue, valueuom → resultunit |
| prescriptions | medicationorders | row_id → recordid, subject_id → patientid, hadm_id → admissionid, starttime → startdatetime, stoptime → enddatetime, drug → medicationname, dose_val_rx → dosevalue, dose_unit_rx → doseunit, route → administrationroute |
| cost | costrecords | row_id → recordid, subject_id → patientid, hadm_id → admissionid, event_type → eventtype, event_id → costid, chargetime → costdatetime, cost → costamount |
| chartevents | clinicalevents | row_id → recordid, subject_id → patientid, hadm_id → admissionid, stay_id → icuadmissionid, itemid → itemcode, charttime → recordeddatetime, valuenum → value, valueuom → unit |
| inputevents | intakerecords | row_id → recordid, subject_id → patientid, hadm_id → admissionid, stay_id → icuadmissionid, starttime → startdatetime, itemid → itemcode, totalamount → totalvolume, totalamountuom → volumeunit |
| outputevents | outputrecords | row_id → recordid, subject_id → patientid, hadm_id → admissionid, stay_id → icuadmissionid, charttime → recordeddatetime, itemid → itemcode, value → volume, valueuom → volumeunit |
| microbiologyevents | microbiologyresults | row_id → recordid, subject_id → patientid, hadm_id → admissionid, charttime → collecteddatetime, spec_type_desc → specimentype, test_name → testname, org_name → organismname |
| icustays | icuepisodes | row_id → recordid, subject_id → patientid, hadm_id → admissionid, stay_id → icuadmissionid, first_careunit → initialcareunit, last_careunit → finalcareunit, intime → admitdatetime, outtime → dischargedatetime |
| transfers | patienttransfers | row_id → recordid, subject_id → patientid, hadm_id → admissionid, transfer_id → transferid, eventtype → transfertype, careunit → careunit, intime → transferindatetime, outtime → transferoutdatetime |

### B.4.2 EICU RENAMING

Table 13: Table and column renaming mappings for eICU.

| Original Table | Mapped Table | Column Mappings (eICU to eICU⋆) |
|---|---|---|
| patient | icupatient | uniquepid → patient_id, patienthealthsystemstayid → hosp_id, patientunitstayid → unit_id, gender → gender, age → age, ethnicity → ethnicity, hospitalid → hospital_id, wardid → ward_id, admissionheight → height_admission, admissionweight → weight_admission, dischargeweight → weight_discharge, hospitaladmittime → hospital_admit_time, hospitaladmitsource → hospital_admission_source, unitadmittime → unit_admit_time, unitdischargetime → unit_discharge_time, hospitaldischargetime → hospital_discharge_time, hospitaldischargestatus → hospital_discharge_status |
| diagnosis | condition | diagnosisid → condition_id, patientunitstayid → unit_id, diagnosisname → condition_name, diagnosistime → condition_time, icd9code → icd9_code |
| treatment | treatment | treatmentid → treatment_id, patientunitstayid → unit_id, treatmentname → treatment_name, treatmenttime → treatment_time |
| lab | lab | labid → lab_id, patientunitstayid → unit_id, labname → lab_name, labresult → lab_result, labresulttime → lab_result_time |
| medication | prescription | medicationid → prescription_id, patientunitstayid → unit_id, drugname → drug_name, dosage → dosage, routeadmin → administration_route, drugstarttime → medication_start_time, drugstoptime → medication_stop_time |
| cost | cost | costid → cost_id, patienthealthsystemstayid → hosp_id, patientunitstayid → unit_id, eventtype → event_type, eventid → event_id, chargetime → cost_time, cost → cost_amount |
| allergy | allergy_reaction | allergyid → allergy_id, patientunitstayid → unit_id, drugname → drug_name, allergyname → allergy_name, allergytime → allergy_time |
| intakeoutput | fluid_balance | intakeoutputid → fluid_balance_id, patientunitstayid → unit_id, cellpath → fluid_path, celllabel → fluid_label, cellvaluenumeric → fluid_value_numeric, intakeoutputtime → fluid_balance_time |
| microlab | microbiology | microlabid → microbiology_id, patientunitstayid → unit_id, culturesite → culture_site, organism → organism, culturetakentime → culture_taken_time |
| vitalperiodic | vital_signs | vitalperiodicid → vital_sign_id, patientunitstayid → unit_id, temperature → temperature, sao2 → sao2, heartrate → heart_rate, respiration → respiration_rate, systemicsystolic → systolic_bp, systemicdiastolic → diastolic_bp, systemicmean → mean_bp, observationtime → vital_time |
| hospital | hospital | hospitalid → hospital_id, numbedscategory → bed_capacity_category, teachingstatus → teaching_status, region → region |

## C  AGENT IMPLEMENTATION

For selecting the backbone LLMs for the agents, small models (e.g., 7B or 13B), DeepSeek-R1, and the Gemma 3 series are excluded due to their limited performance in tool invocation tasks. Owing to budget constraints, we also exclude Anthropic models (e.g., Opus 4 and Sonnet) as well as Gemini-2.5-Pro.

Table 14 presents the system prompt used for our agent baselines. In addition to the system prompt, the agent's input includes three other components: (1) agent behavioral rules detailing interaction behavior (Section C.2); (2) evaluation rules for IncreQA and AdaptQA to ensure accurate evaluation (Section A.6); and (3) database-specific rules outlining the SQL annotation assumptions for MIMIC-IV⋆ and eICU⋆ (Section A.5).

### C.1  AGENT SYSTEM PROMPT

Table 14: Agent system prompt.

```
Instruction:
- You are a DB agent that helps users by answering their questions
in natural language based on information from a database.
- You are currently engaged in a conversation with a user who wants
to retrieve some data or statistics from an EHR database.
- If the user's request is ambiguous or missing details (e.g.,
filtering criteria), ask clarifying questions to better understand
the request.
- You have access to a set of tools to assist the user:
  - table_search: search for tables in the database
  - column_search: search for columns within a table
  - value_substring_search: search for values in a column by
  substring match
  - value_similarity_search: search for values in a column by
  semantic similarity (embedding-based)
  - sql_execute: run SQL queries on the database
  - web_search: retrieve external clinical knowledge not in the
  database
- Use table_search and column_search to explore the database schema.
- Use value_substring_search and value_similarity_search to explore
stored values.
- Clinical concepts (e.g., diagnoses, procedures, medications, lab
tests) in the database may not exactly match the user's words. Use
the value search tools to find relevant entries.
- To search for or verify clinical knowledge not in the database (e.
g., a drug's mechanism of action), use web_search.
- Never invent or assume information that is not provided by the
user or retrieved using the tools.
- Make only one tool call at a time. Do not send a user-facing
response in the same turn as a tool call.
- After gathering all necessary information, use sql_execute to
write and run a single valid SQL query that fully answers the user's
 latest request.
- When you write an SQL query, always execute it with sql_execute
and return the results to the user along with your explanation.

{agent_rule}

{database_rule}
```

### C.2  AGENT RULES

Table 15: Agent rules.

```
Below are the general rules for the DB agent:
```

```
- The DB agent must assume the user has no knowledge of SQL,
databases, or stored values, and cannot execute queries.
- The DB agent must interact with the user only in natural language
and must not show raw SQL queries.
- The DB agent must not modify the database schema or contents. The
following commands are forbidden: INSERT, UPDATE, DELETE, DROP,
ALTER.
- The DB agent must write queries that finish within 60 seconds;
otherwise, the query results will be invalid.
- The DB agent must limit each conversation to 30 interactions (
including user exchanges and tool calls) and 600 seconds total.
- The DB agent must always explain answers in natural language,
including the reasoning or conditions used to arrive at those
answers. If SQL references are necessary, the DB agent must explain
them in terms understandable to someone with no SQL knowledge.
- The DB agent must clearly explain when a question cannot be
answered (e.g., due to limitations of SQL or empty results) and ask
the user to rephrase or modify the request.
- The DB agent must generate a non-empty response, which must
include either a message or a tool call.
```

## C.3  AGENT RULES FOR EVALUATION

Table 16 and Table 17 present the prompts used for IncreQA and AdaptQA tasks, respectively. These prompts are appended to the agent rules to guide the agent toward behavior that aligns with accurate evaluation criteria. Note that for both task types, the agent's responses are evaluated each time they are generated, as correct answers may emerge during the conversation due to the iterative nature of open QA dialogues.

Table 16: Scoring rules for IncreQA.

```
Below are the grading rules:
- The DB agent's performance is evaluated based on the generated SQL
 queries, requiring at least one SQL query (via sql_execute) to
retrieve answers during the interaction with the database.
- For accurate assessment, when the user revises their question, the
 DB agent must write a new SQL query from scratch to fully address
the latest request, without relying on previous query results.
- For questions that involve calculations (such as time differences
or survival rates) or data manipulation/aggregation, the DB agent
must use SQL language to compute the results rather than relying on
its LLM capabilities.
```

Table 17: Scoring rules for AdaptQA.

```
Below are the grading rules:
- The DB agent's performance is evaluated based on the results in
its natural language response to the user.
- When providing answers to the user, the DB agent must enclose the
final answer in <answer></answer> tags (e.g., <answer>42</answer>).
All other content, including intermediate results, explanations,
units, and any additional details, must be placed outside these tags
.
- When answers are textual data (e.g., timestamps or diagnosis names
), use them exactly as stored in the database. For numerical answers
, round them to four decimal places.
```

## D   SAMPLE TRAJECTORIES

In this section, we present sample task instances along with one successful and one failed conversation trajectory for IncreQA and AdaptQA.

### D.1   INCREMENTAL QUERY REFINEMENT

### D.1.1   SUCCESSFUL CASE (INCREQA)

The agent successfully retrieves the number of patients diagnosed with breast cancer by querying values in the description column of the diagnosiscodes table.

**Instruction**: Your goal is to find the number of patients admitted to the hospital who meet specific criteria. Specifically, you want to know how many patients admitted in the past 90 days have a family history of breast cancer.

===

**User**: I'm trying to figure out how many patients admitted to the hospital have certain conditions.

**Agent**: table_search(arguments={"tool_input":""})

**Tool**: admissiondiagnoses, admissionprocedures, clinicalevents, clinicalitemtypes, costrecords, demographics, diagnosiscodes, hospitaladmissions, icuepisodes, intakerecords, labresults, labtesttypes, medicationorders, microbiologyresults, outputrecords, patienttransfers, procedurecodes

**Agent**: column_search(arguments={"table_names":"admissiondiagnoses,diagnosiscodes"})

**Tool**: CREATE TABLE admissiondiagnoses (
        recordid INTEGER NOT NULL,
        patientid INTEGER NOT NULL,
        admissionid INTEGER NOT NULL,
        icdcode TEXT NOT NULL,
        codeversion INTEGER NOT NULL,
        recordeddatetime TEXT NOT NULL,
        PRIMARY KEY (recordid),
        FOREIGN KEY (codeversion) REFERENCES diagnosiscodes (codeversion),
        FOREIGN KEY (icdcode) REFERENCES diagnosiscodes (icdcode),
        FOREIGN KEY (admissionid) REFERENCES hospitaladmissions (admissionid)
)
/*
3 rows from admissiondiagnoses table:
recordid patientid admissionid icdcode codeversion recordeddatetime
0 10035185 22580999 4139 9 2100-05-17 12:53:00

```
1 10035185 22580999 v707 9 2100-05-17 12:53:00
2 10035185 22580999 41401 9 2100-05-17 12:53:00
*/

CREATE TABLE diagnosiscodes (
        recordid INTEGER NOT NULL,
        icdcode TEXT NOT NULL,
        codeversion INTEGER NOT NULL,
        description TEXT NOT NULL,
        PRIMARY KEY (recordid)
)
/*
3 rows from diagnosiscodes table:
recordid icdcode codeversion description
0 0090 9 infectious colitis, enteritis, and gastroenteritis
1 01160 9 tuberculous pneumonia [any form], unspecified
2 01186 9 other specified pulmonary tuberculosis, tubercle bacilli not found by
bacteriological or histological examination, but tuberculosis confirmed by other methods [
inoculation of animals]
*/
```

**Agent**: Could you clarify which specific conditions you'd like to count? For example, are you interested in diabetes, heart failure, COPD, etc.? Also, should these diagnoses be those recorded at the time of hospital admission?

**User**: I want to know how many patients admitted in the last 90 days have a family history of breast cancer.

**Agent**: value_substring_search(arguments={"table":"diagnosiscodes","column":"description","value":"family"})

**Tool**: Values in diagnosiscodes.description containing 'family': ['family history of polycystic kidney', 'non-family member, perpetrator of maltreatment and neglect', 'family history of unspecified malignant neoplasm', ..., 'family history of eye disorders', 'family disruption due to death of family member'].

**Agent**: column_search(arguments={"table_names":"hospitaladmissions"})

**Tool**: CREATE TABLE hospitaladmissions (
        recordid INTEGER NOT NULL,
        patientid INTEGER NOT NULL,
        admissionid INTEGER NOT NULL,

```
        admitdatetime TEXT NOT NULL,
        dischargedatetime TEXT,
        admissiontype TEXT NOT NULL,
        admitsource TEXT NOT NULL,
        dischargedestination TEXT,
        insurancetype TEXT NOT NULL,
        language TEXT,
        maritalstatus TEXT,
        age INTEGER NOT NULL,
        PRIMARY KEY (recordid),
        FOREIGN KEY (patientid) REFERENCES demographics (patientid)
)
/*
3 rows from hospitaladmissions table:
recordid patientid admissionid admitdatetime dischargedatetime admissiontype
admitsource dischargedestination insurancetype language maritalstatus age
0 10004235 24181354 2100-03-19 14:38:00 2100-03-28 14:02:00 urgent transfer from hospital
 skilled nursing facility medicaid english single 47
1 10009628 25926192 2100-09-30 17:08:00 2100-10-08 13:20:00 urgent transfer from
hospital home health care medicaid ? married 58
2 10006053 22942076 2100-11-15 23:39:00 2100-11-17 17:20:00 urgent transfer from hospital
died medicaid english None 52
*/
```

**Agent**: value_substring_search(arguments={"table":"diagnosiscodes","column":"description","value":"breast"})

**Tool**: Values in diagnosiscodes.description containing 'breast': ['other and unspecified disorder of breast associated with childbirth, postpartum condition or complication', 'malignant neoplasm of axillary tail of female breast', 'full-thickness skin loss [third degree, not otherwise specified] of chest wall, excluding breast and nipple', ...', 'personal history of malignant neoplasm of breast', 'open wound of breast, without mention of complication'].

**Agent**: sql_execute(arguments={"query": "SELECT COUNT(DISTINCT ha.patientid) AS num_patients FROM hospitaladmissions AS ha JOIN admissiondiagnoses AS ad ON ha.admissionid = ad.admissionid JOIN diagnosiscodes AS dc ON ad.icdcode = dc.icdcode AND ad.codeversion = dc.codeversion WHERE dc.description = 'family history of malignant neoplasm of breast' AND ha.admitdatetime >= datetime('2100-12-31 23:59:00', '-90 days');"})

**Tool**: [(1,)]

**Agent**: There have been 1 unique patient admitted in the last 90 days with a diagnosis of family history of malignant neoplasm of breast.

**User**: Can you double check that the answer covers my goals?

**Agent**: Yes–here's how the query addresses each part of your goal:
– Last 90 days: We filtered admissions with
ha.admitdatetime $>=$ datetime('2100-12-31 23:59:00','-90 days')
– Family history of breast cancer: We joined admissions -$>$ diagnoses -$>$ codes and used dc.description = 'family history of malignant neoplasm of breast'
– Unique patients: We counted DISTINCT ha.patientid so repeat admissions aren't double-counted.
The result was 1 patient. Let me know if you'd like any further breakdown or checks!

**User**: ###END###

### D.1.2 Failed Case (IncreQA)

The agent fails to retrieve information on breast cancer, as it queries unrelated tables and columns (e.g., clinicalevents and clinicalitemtypes) instead of the diagnosiscodes table. Due to the use of FAISS, wrapped in the LangChain library, values with semantic similarity below the default threshold (0.8) are not retrieved. In the example below, the agent applies the value similarity tool to the itemname column of the clinicalitemtypes table for "cancer," but the tool returns no result. Note that itemname corresponds to the label in the d_item table in MIMIC-IV, which covers a wide range of clinical practices, vital signs, and measurements.

**Instruction**: Your goal is to find the number of patients admitted to the hospital who meet specific criteria. Specifically, you want to know how many patients admitted in the past 90 days have a family history of breast cancer.

===

**User**: I want to find out how many patients have been admitted to the hospital recently and also have a family history of breast cancer.

**Agent**: To assist you with your request, I need to clarify a couple of things:
1. What do you mean by "recently"? Are you looking for admissions within a specific time frame (e.g., the last month, last year)?
2. How is the family history of breast cancer recorded in the database? Is there a specific table or column that indicates this information?
Once I have this information, I can proceed to gather the necessary data from the database
.

**User**: I'm interested in patients admitted in the last 90 days. As for the family history, could you check the database to find where that info is stored?

**Agent**: table_search(arguments={"tool_input":""})

**Tool**: admissiondiagnoses, admissionprocedures, clinicalevents, clinicalitemtypes, costrecords, demographics, diagnosiscodes, hospitaladmissions, icuepisodes, intakerecords, labresults, labtesttypes, medicationorders, microbiologyresults, outputrecords, patienttransfers, procedurecodes

**Agent**: column_search(arguments={"table_names":"demographics"})

**Tool**: CREATE TABLE demographics (
        recordid INTEGER NOT NULL,
        patientid INTEGER NOT NULL,
        gender TEXT NOT NULL,
        dateofbirth TEXT NOT NULL,
        dateofdeath TEXT,

```
        PRIMARY KEY (recordid)
)
/*
3 rows from demographics table:
recordid patientid gender dateofbirth dateofdeath
0 10014729 f 2079-07-22 00:00:00 None
1 10003400 f 2028-07-25 00:00:00 None
2 10002428 f 2020-07-30 00:00:00 None
*/
```

**Agent**: column_search(arguments={"table_names":"clinicalevents"})

```
Tool: CREATE TABLE clinicalevents (
        recordid INTEGER NOT NULL,
        patientid INTEGER NOT NULL,
        admissionid INTEGER NOT NULL,
        icuadmissionid INTEGER NOT NULL,
        itemcode INTEGER NOT NULL,
        recordeddatetime TEXT NOT NULL,
        value DOUBLE PRECISION,
        unit TEXT,
        PRIMARY KEY (recordid),
        FOREIGN KEY (itemcode) REFERENCES clinicalitemtypes (itemcode),
        FOREIGN KEY (icuadmissionid) REFERENCES icuepisodes (icuadmissionid),
        FOREIGN KEY (admissionid) REFERENCES hospitaladmissions (admissionid)
)
/*
3 rows from clinicalevents table:
recordid patientid admissionid icuadmissionid itemcode recordeddatetime value unit
0 10005817 20626031 32604416 220210 2100-12-24 00:00:00 19.0 insp/min
1 10005817 20626031 32604416 220051 2100-12-24 00:00:00 37.0 mmhg
2 10005817 20626031 32604416 223762 2100-12-24 00:00:00 37.0 c
*/
```

**Agent**: column_search(arguments={"table_names":"clinicalitemtypes"})

```
Tool: CREATE TABLE clinicalitemtypes (
        recordid INTEGER NOT NULL,
        itemcode INTEGER NOT NULL,
        itemname TEXT NOT NULL,
        abbreviation TEXT NOT NULL,
        itemtype TEXT NOT NULL,
```

```
        PRIMARY KEY (recordid)
)
/*
3 rows from clinicalitemtypes table:
recordid itemcode itemname abbreviation itemtype
0 226228 gender gender clinicalevents
1 226545 race race clinicalevents
2 229877 suction events (ch) suction events (ch) clinicalevents
*/
```

**Agent**: value_substring_search(arguments={"table":"clinicalitemtypes","column":"itemname","value":"family history"})

**Tool**: No values in clinicalitemtypes.itemname contain 'family history'.

**Agent**: value_similarity_search(arguments={"table":"clinicalitemtypes","column":"itemname","value":"family history"})

**Tool**: No matches found in clinicalitemtypes.itemname for 'family history'.

**Agent**: column_search(arguments={"table_names":"hospitaladmissions"})

**Tool**: CREATE TABLE hospitaladmissions (
```
        recordid INTEGER NOT NULL,
        patientid INTEGER NOT NULL,
        admissionid INTEGER NOT NULL,
        admitdatetime TEXT NOT NULL,
        dischargedatetime TEXT,
        admissiontype TEXT NOT NULL,
        admitsource TEXT NOT NULL,
        dischargedestination TEXT,
        insurancetype TEXT NOT NULL,
        language TEXT,
        maritalstatus TEXT,
        age INTEGER NOT NULL,
        PRIMARY KEY (recordid),
        FOREIGN KEY (patientid) REFERENCES demographics (patientid)
)
/*
3 rows from hospitaladmissions table:
recordid patientid admissionid admitdatetime dischargedatetime admissiontype
admitsource dischargedestination insurancetype language maritalstatus age
```

0 10004235 24181354 2100-03-19 14:38:00 2100-03-28 14:02:00 urgent transfer from hospital
 skilled nursing facility medicaid english single 47
1 10009628 25926192 2100-09-30 17:08:00 2100-10-08 13:20:00 urgent transfer from
hospital home health care medicaid ? married 58
2 10006053 22942076 2100-11-15 23:39:00 2100-11-17 17:20:00 urgent transfer from hospital
died medicaid english None 52
*/

**Agent**: column_search(arguments={"table_names":"demographics"})

**Tool**: CREATE TABLE demographics (
        recordid INTEGER NOT NULL,
        patientid INTEGER NOT NULL,
        gender TEXT NOT NULL,
        dateofbirth TEXT NOT NULL,
        dateofdeath TEXT,
        PRIMARY KEY (recordid)
)
/*
3 rows from demographics table:
recordid patientid gender dateofbirth dateofdeath
0 10014729 f 2079-07-22 00:00:00 None
1 10003400 f 2028-07-25 00:00:00 None
2 10002428 f 2020-07-30 00:00:00 None
*/

**Agent**: column_search(arguments={"table_names":"clinicalevents"})

**Tool**: CREATE TABLE clinicalevents (
        recordid INTEGER NOT NULL,
        patientid INTEGER NOT NULL,
        admissionid INTEGER NOT NULL,
        icuadmissionid INTEGER NOT NULL,
        itemcode INTEGER NOT NULL,
        recordeddatetime TEXT NOT NULL,
        value DOUBLE PRECISION,
        unit TEXT,
        PRIMARY KEY (recordid),
        FOREIGN KEY (itemcode) REFERENCES clinicalitemtypes (itemcode),
        FOREIGN KEY (icuadmissionid) REFERENCES icuepisodes (icuadmissionid),
        FOREIGN KEY (admissionid) REFERENCES hospitaladmissions (admissionid)

```
)
/*
3 rows from clinicalevents table:
recordid patientid admissionid icuadmissionid itemcode recordeddatetime value unit
0 10005817 20626031 32604416 220210 2100-12-24 00:00:00 19.0 insp/min
1 10005817 20626031 32604416 220051 2100-12-24 00:00:00 37.0 mmhg
2 10005817 20626031 32604416 223762 2100-12-24 00:00:00 37.0 c
*/
```

**Agent**: value_substring_search(arguments={"table":"clinicalitemtypes","column":"itemname","value":"cancer"})

**Tool**: No values in clinicalitemtypes.itemname contain 'cancer'.

**Agent**: value_similarity_search(arguments={"table":"clinicalitemtypes","column":"itemname","value":"cancer"})

**Tool**: No matches found in clinicalitemtypes.itemname for 'cancer'.

**Agent**: It seems that I couldn't find any specific entries related to "family history of breast cancer" in the database. This could mean that the information is either stored in a different format or not recorded in the tables I checked.

To proceed, I can look for any general family history or cancer-related entries in the clinical events or demographics tables. Would you like me to do that?

**User**: Yes, please check for any general family history or cancer-related info.

**Agent**: value_substring_search(arguments={"table": "clinicalitemtypes", "column": "itemname", "value": "family history"})

**Tool**: No values in clinicalitemtypes.itemname contain 'family history'.

**Agent**: value_substring_search(arguments={"table":"clinicalitemtypes","column":"itemname","value":"cancer"})

**Tool**: No values in clinicalitemtypes.itemname contain 'cancer'.

**Agent**: I have searched for general family history and cancer-related information in the database, but unfortunately, I couldn't find any relevant entries in the clinical item types. It appears that the information regarding family history of breast cancer may not be explicitly recorded in the database, or it might be stored in a different format or table that I

haven't checked. If you have any other specific criteria or different information you would like to explore, please let me know!

**User**: Okay, can you at least tell me how many patients have been admitted in the last 90 days?

**Agent**: sql_execute(arguments={"query": "SELECT COUNT(DISTINCT patientid) AS patient_count FROM hospitaladmissions WHERE admitdatetime >= '2100-09-30'"})

**Tool**: [(38,)]

**Agent**: There have been 38 patients admitted to the hospital in the last 90 days. If you have any other questions or need further assistance, feel free to ask!

**User**: Okay, so you couldn't find the family history info. Just to confirm, you were able to find the number of patients admitted in the last 90 days?

**Agent**: Yes, that's correct! I was able to find that there have been 38 patients admitted to the hospital in the last 90 days. However, I couldn't find any specific information regarding family history of breast cancer in the database. If you have any other questions or need further assistance, please let me know!

**User**: Double check that you've covered my goals: the number of patients admitted in the last 90 days with a family history of breast cancer.

**Agent**: I understand your request clearly now. You want to know the number of patients who were admitted in the last 90 days and also have a family history of breast cancer. Unfortunately, I was unable to find any specific entries related to family history of breast cancer in the database. This means I cannot confirm how many patients meet both criteria. If you have any other specific information or different criteria you would like to explore, please let me know!

**User**: ###END###

D.2 ADAPTIVE QUERY REFINEMENT

D.2.1 SUCCESSFUL CASE (ADAPTQA)

The user initially asks whether a specific patient was prescribed carbamazepine for epilepsy treatment, but the agent finds that no such records exist for the patient. The user then requests information about other medications for epilepsy. The agent searches the database using clinical knowledge with keywords such as levetiracetam, phenytoin, and phenobarbital. The agent checks whether the patient was prescribed any of these medications and successfully retrieves the most recent prescription date for levetiracetam, as requested.

> **Instruction**: Your goal is to explore patient data related to medication prescriptions. Specifically, you are interested in patient ID 10008287 and want to know when they were prescribed carbamazepine for epilepsy treatment. If the patient was prescribed carbamazepine, you want to know the most recent prescription time after the first epilepsy diagnosis. If the patient was not prescribed carbamazepine, you want to identify whether any other class of medication used for epilepsy treatment was prescribed. If such medications are found, you want to know the most recent prescription time after the first epilepsy diagnosis. If no such medications are found, end the conversation. You want the answer in the exact time format recorded in the database.
>
> ===
>
> **User**: I'm looking into patient prescriptions. Can you help me find some info about the medications prescribed to patient ID 10008287?
>
> **Agent**: table_search(arguments={"tool_input":""})
>
> **Tool**: admissiondiagnoses, admissionprocedures, clinicalevents, clinicalitemtypes, costrecords, demographics, diagnosiscodes, hospitaladmissions, icuepisodes, intakerecords, labresults, labtesttypes, medicationorders, microbiologyresults, outputrecords, patienttransfers, procedurecodes
>
> **Agent**: column_search(arguments={"table_names":"medicationorders"})
>
> **Tool**: CREATE TABLE medicationorders (
>     recordid INTEGER NOT NULL,
>     patientid INTEGER NOT NULL,
>     admissionid INTEGER NOT NULL,
>     startdatetime TEXT NOT NULL,
>     enddatetime TEXT,
>     medicationname TEXT NOT NULL,
>     dosevalue TEXT NOT NULL,
>     doseunit TEXT NOT NULL,
>     administrationroute TEXT NOT NULL,
>     PRIMARY KEY (recordid),

```
        FOREIGN KEY (admissionid) REFERENCES hospitaladmissions (admissionid)
)
/*
3 rows from medicationorders table:
recordid patientid admissionid startdatetime enddatetime medicationname dosevalue
doseunit administrationroute
0 10020740 23831430 2100-04-19 11:00:00 2100-04-20 22:00:00 insulin 0.0 unit sc
1 10005817 20626031 2100-12-20 02:00:00 2100-12-21 12:00:00 insulin 0.0 unit sc
2 10005817 20626031 2100-12-27 10:00:00 2100-12-28 19:00:00 insulin 0.0 unit sc
*/
```

**Agent**: Could you clarify which details you'd like to see for patient 10008287's prescriptions ? For example, are you interested in the medication names only, the start/end dates, dosages, administration routes, or all of those?

**User**: I'm particularly interested in whether patient 10008287 was prescribed carbamazepine for epilepsy. If so, I want to know the most recent time it was prescribed, but only after their first epilepsy diagnosis.

**Agent**: value_substring_search(arguments={"table":"medicationorders","column":"medicationname","value":"carbamazepine","k":10})

**Tool**: Values in medicationorders.medicationname containing 'carbamazepine': ['carbamazepine'].

**Agent**: column_search(arguments={"table_names":"admissiondiagnoses"})

**Tool**: CREATE TABLE admissiondiagnoses (
    recordid INTEGER NOT NULL,
    patientid INTEGER NOT NULL,
    admissionid INTEGER NOT NULL,
    icdcode TEXT NOT NULL,
    codeversion INTEGER NOT NULL,
    recordeddatetime TEXT NOT NULL,
    PRIMARY KEY (recordid),
    FOREIGN KEY (codeversion) REFERENCES diagnosiscodes (codeversion),
    FOREIGN KEY (icdcode) REFERENCES diagnosiscodes (icdcode),
    FOREIGN KEY (admissionid) REFERENCES hospitaladmissions (admissionid)
)
/*
3 rows from admissiondiagnoses table:
recordid patientid admissionid icdcode codeversion recordeddatetime

```
0 10035185 22580999 4139 9 2100-05-17 12:53:00
1 10035185 22580999 V707 9 2100-05-17 12:53:00
2 10035185 22580999 41401 9 2100-05-17 12:53:00
*/
```

**Agent**: column_search(arguments={"table_names":"diagnosiscodes"})

**Tool**: CREATE TABLE diagnosiscodes (
    recordid INTEGER NOT NULL,
    icdcode TEXT NOT NULL,
    codeversion INTEGER NOT NULL,
    description TEXT NOT NULL,
    PRIMARY KEY (recordid)
)
/*
3 rows from diagnosiscodes table:
recordid icdcode codeversion description
0 0090 9 infectious colitis, enteritis, and gastroenteritis
1 01160 9 tuberculous pneumonia [any form], unspecified
2 01186 9 other specified pulmonary tuberculosis, tubercle bacilli not found by
bacteriological or histological examination, but tuberculosis confirmed by other methods [
inoculation of animals]
*/

**Agent**: value_substring_search(arguments={"table":"diagnosiscodes","column":"description","value":"epilepsy","k":10})

**Tool**: Values in diagnosiscodes.description containing 'epilepsy': ['epilepsy complicating pregnancy, childbirth, or the puerperium, delivered, with mention of postpartum complication', 'epilepsy, unspecified, without mention of intractable epilepsy', 'epilepsia partialis continua, with intractable epilepsy', 'epilepsy complicating pregnancy, childbirth, or the puerperium, unspecified as to episode of care or not applicable', 'localization-related (focal) (partial) epilepsy and epileptic syndromes with complex partial seizures, with intractable epilepsy', 'generalized convulsive epilepsy, with intractable epilepsy', 'localization-related (focal) (partial) epilepsy and epileptic syndromes with complex partial seizures, without mention of intractable epilepsy', 'localization-related (focal) (partial) epilepsy and epileptic syndromes with simple partial seizures, without mention of intractable epilepsy', 'epilepsia partialis continua, without mention of intractable epilepsy', 'epilepsy complicating pregnancy, childbirth, or the puerperium, postpartum condition or complication'].

**Agent**: sql_execute(arguments={"query": "SELECT MAX(mo.startdatetime) AS most_recent_prescription FROM medicationorders mo WHERE mo.patientid = 10008287 AND mo.medicationname = 'carbamazepine' AND mo.startdatetime > (SELECT MIN(ad.recordeddatetime) FROM admissiondiagnoses ad JOIN diagnosiscodes dc ON ad.icdcode = dc.icdcode AND ad.codeversion = dc.codeversion WHERE ad.patientid = 10008287 AND dc.description LIKE '%epilepsy%');"})

**Tool**: [(None,)]

**Agent**: sql_execute(arguments={"query": "SELECT ad.recordeddatetime AS diagnosis_time FROM admissiondiagnoses ad JOIN diagnosiscodes dc ON ad.icdcode = dc.icdcode AND ad.codeversion = dc.codeversion WHERE ad.patientid = 10008287 AND dc.description LIKE '%epilepsy%' ORDER BY ad.recordeddatetime ASC LIMIT 1;"})

**Tool**: [('2100-10-09 01:17:00',)]

**Agent**: sql_execute(arguments={"query": "SELECT * FROM medicationorders WHERE patientid = 10008287;"})

**Tool**: [(354, 10008287, 22168393, '2100-10-10 09:00:00', '2100-10-10 10:00:00', 'bag', '1.0', 'bag', 'iv'), (356, 10008287, 22168393, '2100-10-10 07:00:00', '2100-10-10 08:00:00', 'bag', '1.0', 'bag', 'iv'), (655, 10008287, 22168393, '2100-10-09 22:00:00', '2100-10-11 01:00:00', 'cefazolin', '1.0', 'g', 'iv'), ..., (6797, 10008287, 22168393, '2100-10-09 22:00:00', '2100-10-10 10:00:00', '0.9% sodium chloride', '1000.0', 'ml', 'iv'), (6804, 10008287, 22168393, '2100-10-09 22:00:00', '2100-10-10 10:00:00', '0.9% sodium chloride', '1000.0', 'ml', 'iv')]

**Agent**: sql_execute(arguments={"query": "SELECT COUNT(*) AS count_prescriptions FROM medicationorders WHERE patientid = 10008287 AND medicationname = 'carbamazepine';"})

**Tool**: [(0,)]

**Agent**: The answer is <answer>No, patient 10008287 was not prescribed carbamazepine after their first epilepsy diagnosis.</answer>

**User**: Ok, if carbamazepine wasn't prescribed, can you check if they were prescribed any other medications for epilepsy after their first epilepsy diagnosis? If so, give me the most recent prescription time.

**Agent**: sql_execute(arguments={"query": "WITH first_epi AS (SELECT MIN(ad.recordeddatetime) AS dt FROM admissiondiagnoses ad JOIN diagnosiscodes dc ON ad.icdcode = dc.icdcode AND ad.codeversion = dc.codeversion WHERE ad.patientid = 10008287 AND dc.description LIKE '%epilepsy%') SELECT MAX(m.startdatetime) AS

most_recent_prescription FROM medicationorders m JOIN first_epi f ON m.startdatetime > f. dt WHERE m.patientid = 10008287 AND (m.medicationname LIKE '%levetiracetam%' OR m. medicationname LIKE '%phenytoin%' OR m.medicationname LIKE '%phenobarbital%' OR m. medicationname LIKE '%valpro%' OR m.medicationname LIKE '%lamotr%' OR m. medicationname LIKE '%gabapentin%' OR m.medicationname LIKE '%topir%' OR m. medicationname LIKE '%pregabat%' OR m.medicationname LIKE '%ethosux%' OR m. medicationname LIKE '%lacosamide%');"})

**Tool**: [('2100-10-12 20:00:00',)]

**Agent**: The answer is <answer>2100-10-12 20:00:00</answer>

**User**: Great! Just to be sure, this time represents the most recent time patient 10008287 was prescribed any medication for epilepsy after their first diagnosis? I need the answer in the exact time format recorded in the database.

**Agent**: Yes, that is correct. The most recent epilepsy medication prescription time after their first epilepsy diagnosis is <answer>2100-10-12 20:00:00</answer>.

**User**: Perfect, thanks!

###END###

### D.2.2 Failed Case (AdaptQA)

After the user initially asks about carbamazepine for epilepsy treatment for a specific patient, the agent finds no relevant data. The user then asks about other medications for epilepsy. The agent uses a web search tool to retrieve a list of medications, including levetiracetam. The agent writes an SQL query using the diagnosis "epilepsy, unspecified" and the medication list for the patient. However, since the patient was diagnosed with "epilepsy, unspecified, not intractable, without status epilepticus," the agent fails to retrieve the specific data needed, despite having the list of epilepsy medications.

**Instruction**: Your goal is to explore patient data related to medication prescriptions. Specifically, you are interested in patient ID 10008287 and want to know when they were prescribed carbamazepine for epilepsy treatment. If the patient was prescribed carbamazepine, you want to know the most recent prescription time after the first epilepsy diagnosis. If the patient was not prescribed carbamazepine, you want to identify whether any other class of medication used for epilepsy treatment was prescribed. If such medications are found, you want to know the most recent prescription time after the first epilepsy diagnosis. If no such medications are found, end the conversation. You want the answer in the exact time format recorded in the database.

===

**User**: I want to know about the prescriptions for patient ID 10008287.

**Agent**: Could you please specify what information about the prescriptions you are interested in? For example, are you looking for the medication names, dosages, dates, or something else?

**User**: I'm interested in finding out if patient 10008287 was prescribed carbamazepine for epilepsy. If they were, I need the most recent prescription time after their first epilepsy diagnosis.

**Agent**: Okay, I need to find out if patient 10008287 has been diagnosed with epilepsy and prescribed carbamazepine. Then, if both conditions are met, I need to find the most recent prescription time for carbamazepine after the first epilepsy diagnosis. First, I need to find the relevant tables containing diagnoses and medication orders.

**Agent**: table_search(arguments={"tool_input": ""})

**Tool**: admissiondiagnoses, admissionprocedures, clinicalevents, clinicalitemtypes, costrecords, demographics, diagnosiscodes, hospitaladmissions, icuepisodes, intakerecords, labresults, labtesttypes, medicationorders, microbiologyresults, outputrecords, patienttransfers, procedurecodes

**Agent**: Okay, the tables admissiondiagnoses and medicationorders seem relevant. I need to explore these tables to see how diagnoses and medication orders are recorded.

**Agent**: column_search(arguments={"table_names": "admissiondiagnoses, medicationorders "})

**Tool**: CREATE TABLE admissiondiagnoses (
    recordid INTEGER NOT NULL,
    patientid INTEGER NOT NULL,
    admissionid INTEGER NOT NULL,
    icdcode TEXT NOT NULL,
    codeversion INTEGER NOT NULL,
    recordeddatetime TEXT NOT NULL,
    PRIMARY KEY (recordid),
    FOREIGN KEY (codeversion) REFERENCES diagnosiscodes (codeversion),
    FOREIGN KEY (icdcode) REFERENCES diagnosiscodes (icdcode),
    FOREIGN KEY (admissionid) REFERENCES hospitaladmissions (admissionid)
)
/*
3 rows from admissiondiagnoses table:
recordid patientid admissionid icdcode codeversion recordeddatetime
0 10035185 22580999 4139 9 2100-05-17 12:53:00
1 10035185 22580999 v707 9 2100-05-17 12:53:00
2 10035185 22580999 41401 9 2100-05-17 12:53:00
*/

CREATE TABLE medicationorders (
    recordid INTEGER NOT NULL,
    patientid INTEGER NOT NULL,
    admissionid INTEGER NOT NULL,
    startdatetime TEXT NOT NULL,
    enddatetime TEXT,
    medicationname TEXT NOT NULL,
    dosevalue TEXT NOT NULL,
    doseunit TEXT NOT NULL,
    administrationroute TEXT NOT NULL,
    PRIMARY KEY (recordid),
    FOREIGN KEY (admissionid) REFERENCES hospitaladmissions (admissionid)
)
/*
3 rows from medicationorders table:

recordid patientid admissionid startdatetime enddatetime medicationname dosevalue
doseunit administrationroute
0 10020740 23831430 2100-04-19 11:00:00 2100-04-20 22:00:00 insulin 0.0 unit sc
1 10005817 20626031 2100-12-20 02:00:00 2100-12-21 12:00:00 insulin 0.0 unit sc
2 10005817 20626031 2100-12-27 10:00:00 2100-12-28 19:00:00 insulin 0.0 unit sc
*/

**Agent**: Now I need to find the icdcode for epilepsy in the diagnosiscodes table. I will use value_similarity_search to find similar values to "epilepsy" in the description column of the diagnosiscodes table.

**Agent**: value_similarity_search(arguments={"column": "description", "table": "diagnosiscodes", "value": "epilepsy"})

**Tool**: I found close matches in diagnosiscodes.description: ['epilepsy and recurrent seizures ', 'epilepsy, unspecified', 'other epilepsy', 'epileptic spasms', 'epilepsy, unspecified, not intractable', 'epilepsy, unspecified, without mention of intractable epilepsy', 'epilepsy, unspecified, with intractable epilepsy', 'epilepsy, unspecified, intractable', 'other epilepsy and recurrent seizures', 'epilepsy, unspecified, intractable, without status epilepticus', ' epilepsy, unspecified, not intractable, without status epilepticus', 'other epilepsy, intractable, without status epilepticus', 'epilepsy, unspecified, intractable, with status epilepticus', 'other epilepsy, intractable, with status epilepticus', 'epilepsy, unspecified, not intractable, with status epilepticus', 'generalized idiopathic epilepsy and epileptic syndromes', 'other epilepsy, not intractable, without status epilepticus', 'generalized idiopathic epilepsy and epileptic syndromes, intractable'].

**Agent**: The value similarity search returned several matches. To proceed, I will use 'epilepsy , unspecified' as the diagnosis for epilepsy. Now I need to check if patient 10008287 was diagnosed with 'epilepsy, unspecified' and prescribed 'carbamazepine'. If so, I need to find the most recent prescription time after the first epilepsy diagnosis.

**Agent**: value_substring_search(arguments={"column": "medicationname", "table": "medicationorders", "value": "carbamazepine"})

**Tool**: Values in medicationorders.medicationname containing 'carbamazepine': [' carbamazepine'].

**Agent**: Okay, the medication name is 'carbamazepine'. Now I have all the information I need to construct the SQL query.

**Agent**: sql_execute(arguments={"query": "SELECT MAX(m.startdatetime) FROM medicationorders m JOIN admissiondiagnoses a ON m.patientid = a.patientid WHERE m.

patientid = 10008287 AND m.medicationname = 'carbamazepine' AND a.icdcode IN (SELECT icdcode FROM diagnosiscodes WHERE description = 'epilepsy, unspecified') AND m. startdatetime > (SELECT MIN(recordeddatetime) FROM admissiondiagnoses a2 WHERE a2. patientid = 10008287 AND a2.icdcode IN (SELECT icdcode FROM diagnosiscodes WHERE description = 'epilepsy, unspecified'))"})

**Tool**: [(None,)]

**Agent**: The query returned None, which means either the patient was not diagnosed with epilepsy, was not prescribed carbamazepine, or the prescription occurred before the diagnosis.

**Agent**: sql_execute(arguments={"query": "SELECT MIN(recordeddatetime) FROM admissiondiagnoses a WHERE a.patientid = 10008287 AND a.icdcode IN (SELECT icdcode FROM diagnosiscodes WHERE description = 'epilepsy, unspecified')"})

**Tool**: [(None,)]

**Agent**: Since the query to find the first epilepsy diagnosis returned None, it means the patient was never diagnosed with epilepsy. Therefore, I cannot find the most recent carbamazepine prescription time after the first epilepsy diagnosis.

**Agent**: sql_execute(arguments={"query": "SELECT DISTINCT patientid FROM medicationorders WHERE medicationname = 'carbamazepine' AND patientid = 10008287"})

**Tool**: []

**Agent**: The patient 10008287 was not prescribed carbamazepine. Therefore, I cannot find the most recent carbamazepine prescription time after the first epilepsy diagnosis. < answer>Patient 10008287 was not diagnosed with epilepsy and was not prescribed carbamazepine.</answer>

**User**: Okay, if they weren't prescribed carbamazepine, I'd like to know if they were prescribed any other medications for epilepsy. If so, I want the most recent prescription time after their first epilepsy diagnosis, just like I asked for carbamazepine.

**Agent**: Okay, I need to find out if patient 10008287 has been diagnosed with epilepsy. If they have, I need to find any medications prescribed to them that are used to treat epilepsy and then find the most recent prescription time for those medications after the first epilepsy diagnosis. First, I need to identify medications used to treat epilepsy. I will use a web search for this.

**Agent**: web_search(arguments={"query": "medications used to treat epilepsy"})

**Tool**: [
"Below is a list of some of the most common brand-name drugs currently used to treat epilepsy. Your doctor may prefer that you take the brand name of anticonvulsant and not the generic substitution. Talk with your doctor about this important issue.\n\nBrivaracetam (Briviact)\n\nCannabidiol (Epidiolex)\n\nCarbamazepine (Carbatrol or Tegretol)\n\nCenobamate(Xcopri)\n\nDiazepam (Valium), lorazepam (Ativan) and similar benzodiazepine tranquilizers such as clonazepam (Klonopin)\n\nEslicarbazepine (Aptiom) [...] Ethosuximide (Zarontin)\n\nFelbamate (Felbatol)\n\nFenfluramine (Fintepla)\n\nLacosamide (VIMPAT)\n\nLamotrigine (Lamictal)\n\nLevetiracetam (Keppra)\n\nOxcarbazepine (Oxtellar XR, Trileptal)\n\nPerampanel (Fycompa)\n\nPhenobarbitol\n\nPhenytoin (Dilantin)\n\nPregabalin (Lyrica)\n\nTiagabine (Gabitril)\n\nTopiramate (Topamax)\n\nValproate, valproic acid (Depakene, Depakote)\n\nZonisamide (Zonegran) [...] The most commonly used medications are benzodiazepines because they get into the bloodstream quickly to start working on the brain to stop the seizure. They include:\n\n## Epilepsy Drug Guidelines\n\nIt may take several months before the best drug and dosage are determined for you. During this adjustment period, you will be carefully monitored through frequent blood tests to measure your response to the medication.",
"Below is a list of some of the most common brand-name drugs currently used to treat epilepsy. Your doctor may prefer that you take the brand name of anticonvulsant and not the generic substitution. Talk with your doctor about this important issue.\n\nBrivaracetam (Briviact)\n\nCannabidiol (Epidiolex)\n\nCarbamazepine (Carbatrol or Tegretol)\n\nCenobamate(Xcopri)\n\nDiazepam (Valium), lorazepam (Ativan) and similar benzodiazepine tranquilizers such as clonazepam (Klonopin)\n\nEslicarbazepine (Aptiom) [...] Ethosuximide (Zarontin)\n\nFelbamate (Felbatol)\n\nFenfluramine (Fintepla)\n\nLacosamide (VIMPAT)\n\nLamotrigine (Lamictal)\n\nLevetiracetam (Keppra)\n\nOxcarbazepine (Oxtellar XR, Trileptal)\n\nPerampanel (Fycompa)\n\nPhenobarbitol\n\nPhenytoin (Dilantin)\n\nPregabalin (Lyrica)\n\nTiagabine (Gabitril)\n\nTopiramate (Topamax)\n\nValproate, valproic acid (Depakene, Depakote)\n\nZonisamide (Zonegran) [...] The most commonly used medications are benzodiazepines because they get into the bloodstream quickly to start working on the brain to stop the seizure. They include:\n\n## Epilepsy Drug Guidelines\n\nIt may take several months before the best drug and dosage are determined for you. During this adjustment period, you will be carefully monitored through frequent blood tests to measure your response to the medication.",
"Medications used to treat epilepsy are called antiepileptic drugs (AEDs). There are more than 30 prescription AEDs on the market, and they're mostly available as oral tablets or capsules.\n\nYour options depend on:\n\n## Types of antiepileptic drugs (AEDs)\n\nThere are two types of medications for epilepsy:\n\nSome people may need to take more than one medication to prevent seizures. [...] ### Levetiracetam (Elepsia XR, Keppra, Keppra XR, Spritam)\n\nLevetiracetam (Elepsia XR, Keppra, Keppra XR, Spritam) may treat a range of epileptic seizures, including:\n\nIt's available as a pill, an IV solution, an

oral solution, and an injection.\n\nLevetiracetam may cause fewer side effects than other medications used for epilepsy. It's safe to take during pregnancy, according to experts such as the United Kingdom's Commission on Human Medicines.\n\n### Lorazepam (Ativan) [...] Lorazepam (Ativan) is a benzodiazepine that's used to treat all types of seizures. It's also used to treat status epilepticus. Status epilepcticus is a prolonged, critical seizure that's regarded as a medical emergency.\n\nIt's available as a pill, an oral concentrate, and an injection.\n\n### Methsuximide (Celontin)\n\nMethsuximide (Celontin) is used for absence seizures. It's prescribed when other treatments don't work in treating your seizures.",

"Donate\n\nPopular searches: Diagnosing Epilepsy Treatments and Therapies what is epilepsy\n\nMake an Impact\n\nEpilepsy and Seizure Medications\n ===============================\n\n#### Learn about FDA-approved medications to treat epilepsy and seizures.\n\nImage 7: Epilepsy and Seizure Medications\n\nSearch\n\nFilter Alphabetically Select Clear\n\nLoading ...\n\n##### Acetazolamide\n\nMore info( Brand names: Diamox, Diamox Sequels, generics) [...] Acetazolamide (a SEET a ZOLE a mide) has been FDA-approved for the treatment of, along with other drugs, centrencephalic epilepsies (absence, generalized seizures).\n\n##### Brivaracetam\n\nMore info(Brand names: Briviact)\n\nBrivaracetam (briv a RA se tam) has been approved by the FDA to treat focal (partial) onset seizures in patients 1 month of age and older.\n\n##### Cannibadiol\n\nMore info(Brand names: Epidiolex) [...] More info(Brand names: Aptiom)\n\nEslicarbazepine acetate (ES li Kar BAZ e peen) has been approved by the FDA to treat focal onset seizures in patients 4 years of age and older.\n\n##### Ethosuximide\n\nMore info(Brand names: Zarontin, generics)\n\nEthosuximide (ETH oh SUX i mide) has been approved by the FDA to control absence (petit mal) epilepsy.\n\n ##### Ethotoin\n\nMore info(Brand names: Peganone, generics)",

"Healthcare providers prescribe antiseizure medications to treat epilepsy and symptomatic seizures. They also prescribe these medications to prevent and/or treat seizures that happen during or following brain surgery. [...] Gray gradient\nGray gradient\ nSearch Icon Blue\nCleveland Clinic logo\n\n# Antiseizure Medications (Formerly Known as Anticonvulsants)\n\nAntiseizure medications (anticonvulsants) help treat epilepsy and other causes of seizures. They can treat other conditions as well, like anxiety and neuropathic pain. There are several different types of antiseizure medications. You and your healthcare provider will work together to find the best one for you.\n\ nAdvertisement [...] Cleveland Clinic is a non-profit academic medical center. Advertising on our site helps support our mission. We do not endorse non-Cleveland Clinic products or services. Policy\n\n## Overview\n\n### What are antiseizure medications ( anticonvulsants)?\n\nAntiseizure medications (previously known as antiepileptic or anticonvulsant medications) are prescription medications that help treat and prevent seizures. Healthcare providers may prescribe these medications to treat other conditions as well.",

"Epilepsy Website Logo\n\n# List of Anti-Seizure Medications (ASMs)\n\n#### Understanding Epilepsy\n\nAnti-epileptic drugs (ASMs) are the main form of treatment

for people living with epilepsy, with up to 70% (7 in 10 people) having their seizures controlled through this medication.\n\nIn Australia there are over 20 ASMsare used to treat seizures. The ASMsprescribed are often selected on the basis of the seizure type/s, age, gender and side effects. ASMsmay be prescribed as tablets, syrups and liquids.",

"One large randomized trial, the Standard and New Antiepileptic Drugs (SANAD) trial, demonstrated some comparative advantages of certain AEDs when treating focal or generalized epilepsy. In the end, when comparing valproate, lamotrigine, or topiramate for generalized seizures, they recommended valproic acid as their first-line choice. Additionally, when comparing carbamazepine, gabapentin, lamotrigine, oxcarbazepine, and topiramate for focal seizures, lamotrigine was cited as the first-line [...] In summary, it is now abundantly clear that anti-seizure medications wield disparate mechanistic profiles, but they all effectively suppress epileptic seizures in one way or another. Accordingly, grouping the drugs together by mechanism is a very helpful organizing principle. From this viewpoint, it may become easier to appreciate that some drugs have different efficacy profiles for different seizures types and epilepsy syndromes. Ethosuximide is an exception with its specific limited use with [...] Phenytoin is one of the oldest anti-seizure medications and is still widely used for focal and generalized seizures. It is also administered for status epilepticus. In addition, practitioners may invoke phenytoin as a second-line agent for patients with mixed seizure types (e.g., tonic-clonic and myoclonic). As mentioned, phenytoin blocks voltage-gated sodium channels, but other possible mechanisms revolve around decreased synaptic transmission, smaller changes in ionic gradients involving the",

"Medicines.\n Surgery.\n Therapies that stimulate the brain using a device.\n A ketogenic diet.\n\n### Medication\n\nMost people with epilepsy can become seizure-free by taking one anti-seizure medicine, which is also called an anti-epileptic medicine. Others may be able to decrease the number and intensity of their seizures by taking more than one medicine. [...] Tell your healthcare professional immediately if you notice new or increased feelings of depression or suicidal thoughts. Also contact your healthcare professional right away if you have changes in your mood or behaviors.\n Tell your healthcare professional if you have migraines. You may need an anti-seizure medicine that can prevent your migraines and treat epilepsy. [...] ### Potential future treatments\n\nResearchers are studying many potential new treatments for epilepsy, including:\n\n Continuous stimulation of the seizure onset zone, known as subthreshold stimulation. Subthreshold stimulation is continuous stimulation to an area of the brain below a level that's physically noticeable. This type of therapy appears to improve seizure outcomes and quality of life for some people with seizures. Subthreshold stimulation helps stop a seizure before it happens.",

"| stiripentol | Diacomite | Image 50: Medline unavailable | Image 51: Medline unavailable |\n| sulthiame | Ospolot | Image 52: Medline available | Image 53: Medline unavailable |\n | tiagabine | Gabitril | Image 54: Medline available | Image 55: Medline available |\n| topiramate | Topamax | Image 56: Medline available | Image 57: Medline available |\n| vigabatrin | Sabril | Image 58: Medline available | Image 59: Medline available | [...] |

ethosuximide | Zarontin | Image 20: Medline available | Image 21: Medline available |\n|
felbamate | Felbatol | Image 22: Mims unavailable | Image 23: Medline available |\n|
gabapentin | Neurontin | Image 24: Medline available | Image 25: Medline available |\n|
lacosamide | Vimpat | Image 26: Medline available | Image 27: Medline available |\n|
lamotrigine | Lamictal | Image 28: Medline available | Image 29: Medline available | [...] |
phenobarbitone | Phenobarbitone | Image 40: Medline available | Image 41: Medline
unavailable |\n| phenytoin | Dilantin | Image 42: Medline available | Image 43: Medline
available |\n| potassium bromide | Dibro-Be mono | Image 44: Medline unavailable |
Image 45: Medline unavailable |\n| primidone | Mysoline | Image 46: Medline available |
Image 47: Medline available |\n| sodium valproate | Epilim | Image 48: Medline available |
Image 49: Medline available |",
"Sodium valproate is an epilepsy drug prescribed for all seizure types including absence,
myoclonus and tonic clonic seizures. New regulations have been introduced by the
Medicines and Healthcare products Regulatory Agency (MHRA) around the way in which
the drug, sodium valproate is prescribed to women and girls of childbearing age.\n\nLET'
S STAY IN TOUCH... [...] The decision whether to start taking anti-epileptic drugs (AEDs)
can be difficult, and there is a lot to think about. Here we look at the benefits and risks of
taking, or not taking, AEDs.\n\nSodium valproate [...] Epilepsy Society\n\nSite
Navigation\n\nUtility Nav\n\nSite Search\n\nbreadcrumb navigation:\n\nList of anti-
seizure medication\n\nPublished on 8 October 2024\n\nUpdated: 7 April 2025\n\
nAuthored by Anonymous\n\nAlso in this section\n\nList of anti-seizure medication\n\
nHere is a list of anti-seizure medication (ASM), previously called anti-epileptic drugs or
AEDs."
]

**Agent**: Based on the web search results, here's a list of medications commonly used to
treat epilepsy:
Brivaracetam
Cannabidiol
Carbamazepine
Cenobamate
Diazepam
Lorazepam
Clonazepam
Eslicarbazepine
Ethosuximide
Felbamate
Fenfluramine
Lacosamide
Lamotrigine
Levetiracetam
Oxcarbazepine

Perampanel

Phenobarbitol

Phenytoin

Pregabalin

Tiagabine

Topiramate

Valproate, valproic acid

Zonisamide

Acetazolamide

Methsuximide

Stiripentol

Sulthiame

Gabapentin

Primidone

Sodium valproate

I will now construct a query to find the most recent prescription time for any of these medications after the first epilepsy diagnosis.

**Agent**: sql_execute(arguments={"query": "SELECT MAX(m.startdatetime) FROM medicationorders m JOIN admissiondiagnoses a ON m.patientid = a.patientid WHERE m.patientid = 10008287 AND m.medicationname IN ('brivaracetam', 'cannabidiol', 'carbamazepine', 'cenobamate', 'diazepam', 'lorazepam', 'clonazepam', 'eslicarbazepine', 'ethosuximide', 'felbamate', 'fenfluramine', 'lacosamide', 'lamotrigine', 'levetiracetam', 'oxcarbazepine', 'perampanel', 'phenobarbitol', 'phenytoin', 'pregabalin', 'tiagabine', 'topiramate', 'valproate', 'valproic acid', 'zonisamide', 'acetazolamide', 'methsuximide', 'stiripentol', 'sulthiame', 'gabapentin', 'primidone', 'sodium valproate') AND a.icdcode IN ( SELECT icdcode FROM diagnosiscodes WHERE description = 'epilepsy, unspecified') AND m.startdatetime > (SELECT MIN(recordeddatetime) FROM admissiondiagnoses a2 WHERE a2.patientid = 10008287 AND a2.icdcode IN (SELECT icdcode FROM diagnosiscodes WHERE description = 'epilepsy, unspecified'))"})

**Tool**: [(None,)]

**Agent**: The query returned None, which means either the patient was not diagnosed with epilepsy, was not prescribed any of the listed medications, or the prescriptions occurred before the diagnosis. Since I already determined that the patient was not diagnosed with epilepsy, the result is None. <answer>Patient 10008287 was not diagnosed with epilepsy and was not prescribed any anti-epileptic medications.</answer>

**User**: Could you double check that your answer covers everything? I wanted to know if patient 10008287 was prescribed carbamazepine and, if not, if they were prescribed any

other epilepsy meds. For any epilepsy meds, I need the most recent prescription time after their first epilepsy diagnosis.

**Agent**: Yes, I have confirmed that patient 10008287 was neither diagnosed with epilepsy nor prescribed carbamazepine or any other anti-epileptic medications from the list I generated using a web search. Therefore, I cannot provide a prescription time after their first epilepsy diagnosis. <answer>Patient 10008287 was not diagnosed with epilepsy and was not prescribed carbamazepine or any other anti-epileptic medications.</answer>

**User**: ###END###

### D.3 Invalidated Trajectories by the Validator

Below we show sample trajectories that were invalidated by our simulation validator along with the specific rules they violated.

#### D.3.1 Violation of No Final Check Rule

**Instruction**: Your goal is to find the administration routes of specific ordered medications. Specifically, you are interested in the administration routes of ordered medications containing ISDN. You want to search for isosorbide dinitrate specifically, not other similar medications like \"isosorbide mononitrate\" When querying the DB agent, since you do not know how isosorbide dinitrate is stored, use common terms like \"ISDN\" when referring to it, and let the DB agent find it for you.

===

**User**: I'm trying to figure out how certain meds are given to patients. Can you help?

**Agent**: I can help with that. Could you please specify which medications you are interested in and what information about their administration you would like to know? For example, are you interested in the dosage, frequency, or route of administration?

(...omitted...)

**Agent**: The administration route for isosorbide dinitrate is 'po/ng'. This likely refers to oral or nasogastric administration.

**User**: ###END### (RULE VIOLATION)

→ *Before ending the conversation, ask the DB agent to double check that their final answer covers all goals raised. If not, request any missing parts.*

#### D.3.2 Violation of Missing conditions Rule

**Instruction**: Your goal is to find information related to a patient's diagnosis. Specifically, you are interested in patient ID 10004733 and you want to know how many days have passed since the patient was diagnosed with cerebral artery occlusion during the current hospital admission. If the patient has multiple records of the same diagnosis, you want to know the number of days elapsed since the earliest diagnosis. You want diagnosis names containing cerebral artery occlusion.

===

**User**: I'm trying to find out some information about a patient's diagnosis.

**Agent**: I can help with that. What specific information are you looking for regarding a patient's diagnosis? For example, are you trying to find:
– A patient's diagnosis history?
– The most common diagnoses?
– Diagnoses associated with certain procedures or medications?
– The cost associated with a particular diagnosis?
– Something else entirely?
Please provide more details so I can narrow down my search.

**User**: I'm interested in patient ID 10004733. I want to know how long ago they were diagnosed with something.

**Agent**: Okay, I can help you find out how long ago patient ID 10004733 was diagnosed. Let me first check the admissiondiagnoses table to see if it contains the diagnosis date and patient ID.

**User**: ###END### (RULE VIOLATION)

→ *The user must provide all detailed conditions specified in the instruction before ending the conversation.*

### D.3.3  VIOLATION OF PERFORMING AGENT-LIKE TASKS RULE

**Instruction**: Your goal is to find the number of blood culture tests performed on a specific patient. Specifically, you want to know how many culture tests were collected from venipuncture blood for patient ID 031–10300 this month.

===

**User**: I need to figure out how many blood culture tests a particular patient has had.

**Agent**: To determine the number of blood culture tests, I need the patient's ID. Could you please provide the patient ID?

**User**: The patient ID is 031–10300.

(…omitted…)

**Agent**: The patient with ID 031–10300 had 2 blood culture tests collected from venipuncture blood.

**User**: Okay, I have the patient ID, the time frame (this month), and the collection method (venipuncture blood). Does your request include any other conditions? (RULE VIOLATION)

→ *The user acts like a DB agent or AI assistant instead of the user (e.g., writing, reviewing, or executing SQL queries, calling external APIs, or responding to the DB agent in a machine assistant way).*

