# OpenReview forum: "From Conversation to Query Execution: Benchmarking User and Tool Interactions for EHR Database Agents"
_ICLR.cc/2026/Conference — ICLR 2026 Poster_

### Official Review · Reviewer_Rw7P · 2025-10-31

**Soundness:** 3
**Presentation:** 3
**Contribution:** 3
**Rating:** 4
**Confidence:** 3

**Summary:**

The paper introduces EHR-ChatQA, a new interactive benchmark for evaluating Large Language Model (LLM) agents accessing Electronic Health Record (EHR) databases, specifically addressing the practical challenges of query ambiguity and value mismatch common in clinical settings. EHR-ChatQA assesses the full end-to-end agentic workflow, including conversational refinement using an LLM-based user simulator, active schema and value tool use, and accurate SQL generation, across two distinct interactive flows: Incremental Query Refinement (IncreQA) and the more adaptive AdaptQA. Evaluations conducted across state-of-the-art closed- and open-source models reveal a critical robustness gap (Pass@5 versus Pass^5), exceeding 35-60% in the challenging AdaptQA flow, demonstrating that current agents, while occasionally capable, are fundamentally unreliable for safety-critical EHR tasks.

**Strengths:**

- The benchmark is the first interactive evaluation specifically designed for EHR question answering, holistically assessing the full agentic workflow including tool use and conversational refinement.
- Construction of the task instances is rigorously grounded in real-world clinical QA scenarios, utilizing publicly available EHR databases with renamed schemas to enforce genuine exploration rather than memorization.
- A sophisticated simulation environment employs a stochastic LLM user and a dedicated LLM-as-a-judge validator to mitigate simulation noise and ensure evaluation fidelity.
- Evaluation metrics centered on consistent success (Pass^k) and the resulting robustness gap (Gap-k) provide vital diagnostic insights into agent reliability, critical for safety-sensitive applications.

**Weaknesses:**

- The Adaptive Query Refinement (AdaptQA) flow, which represents the more challenging and novel adaptation scenario, comprises only 64 instances, potentially limiting the statistical robustness of evaluations in this critical area.
- Although the user is simulated by an LLM, its conversational spontaneity is heavily constrained by strict behavioral rules (Table 5), raising questions about the fidelity of the simulation to real clinical dialogue complexity.
- Evaluation relies significantly on extensive, database-specific SQL generation rules (Tables 8 and 9) provided to the agent, suggesting the benchmark may test adherence to prompting constraints rather than intrinsic database reasoning ability.
- Performance of open-source models on AdaptQA is critically low (0.0% Pass^5), which diminishes the utility of the benchmark for diagnosing failures in current open LLM architectures.
- More quantitative diagnostic insights are needed, detailing which specific types of user utterances or tool outputs most frequently trigger the observed dramatic drop in consistent success (Gap-k).
- Lack of transparency regarding the hyperparameters and thresholds used for the `value_similarity_search` tool makes replication or external validation of agent interaction logic difficult.

**Questions:**

- Given the importance of AdaptQA in revealing the robustness gap, what efforts are planned to expand its size to improve the statistical reliability of results for this difficult and novel task type?
- Could authors provide a more detailed, quantitative breakdown of the error types (Value Linking vs. SQL Generation) specifically correlated with the Gap-k metric, differentiating brittle failures from consistent failures?
- Since simulation validity is crucial, please provide a few detailed examples of dialogues deemed "invalid" by the LLM-as-a-judge validator to better illustrate how the validator enforces the specified user behavior rules.

---

> ### Author Response · Authors · 2025-11-24
>
> We thank the reviewer for the constructive feedback and detailed assessment. Please find our responses below.
>
> **[W1 & Q1] The Adaptive Query Refinement (AdaptQA) flow, which represents the more challenging and novel adaptation scenario, comprises only 64 instances, potentially limiting the statistical robustness of evaluations in this critical area. Given the importance of AdaptQA in revealing the robustness gap, what efforts are planned to expand its size to improve the statistical reliability of results for this difficult and novel task type?**
>
> We appreciate the reviewer’s constructive suggestion. We agree that expanding the AdaptQA dataset will further strengthen the statistical robustness of evaluations in this critical area. Below, we briefly explain the inherent challenges in constructing AdaptQA tasks and outline our concrete plan for expansion.
> - Complexity of Task Construction (Quality over Quantity): Our design philosophy prioritized clinical realism and logic-driven query adaptation over raw volume. As shown in Table 13, AdaptQA comprises diverse categories grounded in reasonable fallback logic (e.g., data absence) rather than arbitrary preference changes. This involves a “dual constraint" challenge for constructing these tasks: (1) a valid and reasonable clinical adaptation scenario (not due to simple preference changes), and (2) the existence of specific patient patterns in the database. As a result, locating such specific "absence-presence" patterns within the MIMIC-IV and eICU databases requires extensive manual exploration as well as checking the clinical soundness of the task scenarios.
> - Expansion Progress and Plan: That being said, we agree that increasing the dataset size is beneficial. We have already initiated the expansion and plan to release 16 additional high-quality tasks before the end of the discussion period. This will bring the total AdaptQA dataset to 80 instances (64 + 16), an increase of 25%. Subsequently, we commit to conducting the evaluation on these new samples and updating the experimental results prior to the paper decision notification.
>
> **[W2] Although the user is simulated by an LLM, its conversational spontaneity is heavily constrained by strict behavioral rules (Table 5), raising questions about the fidelity of the simulation to real clinical dialogue complexity.**
>
> We would like to clarify that these rules are introduced for evaluation reproducibility and validity. In an uncontrolled simulation with only the task instruction given, spontaneity often leads to hallucinated questions not mentioned in the instruction, conversations that never end, or SQL or AI assistant-like outputs that resemble LLM behavior rather than that of humans who do not know SQL or database systems and want help from the agent. These constraints help the simulated user follow the task instruction more faithfully and in a more human-like way without deviating from our intended user-agent dialogue complexity.
>
> To better illustrate the resulting conversational variability, we added a new section in Appendix A.3.3 (Example Variability in Simulated User Turns), showing the variability in user turns per instruction. For each task, we compute the difference between the maximum and minimum number of user turns across trials. In IncreQA, 238 out of 286 tasks (about 83 percent) have a difference of 1 to 3 turns, and in AdaptQA, 50 out of 64 tasks (about 78 percent) also fall in this range. These distributions show that the number of turns taken by the simulated user varies across different trajectories under the same instruction.

---

> > ### Author Response · Authors · 2025-11-25
> >
> > **[W3] Evaluation relies significantly on extensive, database-specific SQL generation rules (Tables 8 and 9) provided to the agent, suggesting the benchmark may test adherence to prompting constraints rather than intrinsic database reasoning ability.**
> >
> > We respectfully argue that providing database-specific rules is not a prompting constraint but a necessity for realistic EHR agents and fair evaluation. Real-world EHR schemas (like MIMIC-IV and eICU) are highly idiosyncratic and complex. These rules serve two critical purposes:
> > - Reflecting Real-world Deployment: As far as we understand, in most production-level text-to-SQL systems, providing a data dictionary (i.e., schema definitions) and standardized business logic (e.g., specific calculation rules) is standard engineering practice. Without these explicit guidelines, even human experts often fail to query databases correctly because conventions vary by institution. Our benchmark evaluates the agent's ability to use these standard manuals to solve problems rather than its ability to guess hidden database conventions.
> > - Standardized Evaluation: Complex questions often allow for multiple syntactically valid SQL queries that yield slightly different results. For example, a "ranking" request could be implemented using either RANK() or DENSE_RANK(). Without explicit rules enforcing one standard, an agent might be penalized for a valid stylistic choice that differs from the ground truth.
> >
> > Crucially, despite these constraints, our benchmark still rigorously tests the agent's autonomy to invoke tools for schema exploration, ask clarifying questions to resolve ambiguity, resolve value mismatches (mapping user terms to database values), and correctly generate SQL based on the schema context.
> >
> > **[W4] Performance of open-source models on AdaptQA is critically low (0.0% Pass^5), which diminishes the utility of the benchmark for diagnosing failures in current open LLM architectures.**
> >
> > We believe this finding is a crucial contribution of our benchmark rather than a limitation. The fact that open-source models achieve a non-zero Pass@5 (optimistic success) but a near-zero Pass^5 (consistent success) empirically validates the existence of the "Robustness Gap" we introduce. This highlights that current open-source models are brittle in maintaining context over complex and long context horizons (AdaptQA), identifying a specific area for future model development.  We believe that good benchmarks are the ones that can differentiate between capable models from the models that are not, and our work successfully distinguishes state-of-the-art closed-source models from current open-source alternatives.
> >
> > **[W5 & Q2] More quantitative diagnostic insights are needed, detailing which specific types of user utterances or tool outputs most frequently trigger the observed dramatic drop in consistent success (Gap-k). Could authors provide a more detailed, quantitative breakdown of the error types (Value Linking vs. SQL Generation) specifically correlated with the Gap-k metric, differentiating brittle failures from consistent failures?**
> >
> > In Section 5.3, we provide a quantitative breakdown of "Inconsistent Failures," which constitute the primary driver of the Gap-k metric. Our analysis reveals that 71.8% of these inconsistent failures stem from SQL generation issues within the conversational context, significantly outweighing value linking errors (15.4%). This stands in sharp contrast to "Consistent Failures," which are primarily driven by value linking errors (46.9%). Regarding the specific triggers for the gap, we identify two primary failure modes: (1) Context Loss, where agents fail to carry over constraints from previous turns when user phrasing varies slightly (e.g., selecting “all patients” when the context specifies "patients with diabetes"), and (2) Semantic Errors, where agents generate logically incorrect SQL, such as confusing patient_id with admission_id or misinterpreting event sequences, which also appear in consistent failure cases. Consequently, we conclude that context-dependent text-to-SQL generation is the primary bottleneck causing the drop in consistent success (Gap-k), rather than the agent’s lack of schema or value exploration. If you have specific areas you would like us to examine beyond our analyses, please feel free to add comments.

---

> > > ### Author Response · Authors · 2025-11-25
> > >
> > > **[Q3] Since simulation validity is crucial, please provide a few detailed examples of dialogues deemed "invalid" by the LLM-as-a-judge validator to better illustrate how the validator enforces the specified user behavior rules.**
> > >
> > > We agree that providing concrete examples is essential to illustrate the validity of our simulation validator. While Section 6 summarizes the error distribution, we present below the three most frequent error types caught by our validator. These examples have also been included in Appendix D.3.
> > >
> > > 1: Violation of "No Final Check" Rule
> > >
> > > - User Instruction: Your goal is to find the administration routes of specific ordered medications. Specifically, you are interested in the administration routes of ordered medications containing ISDN. You want to search for isosorbide dinitrate specifically, not other similar medications like \"isosorbide mononitrate\" When querying the DB agent, since you do not know how isosorbide dinitrate is stored, use common terms like \"ISDN\" when referring to it, and let the DB agent find it for you.
> > >
> > > - Conversation History:
> > >
> > >   - User: I'm trying to figure out how certain meds are given to patients. Can you help?
> > >   - Agent: I can help with that. Could you please specify which medications you are interested in and what information about their administration you would like to know? For example, are you interested in the dosage, frequency, or route of administration?
> > >   - (...omitted…)
> > >   - Agent: The administration route for isosorbide dinitrate is 'po/ng'. This likely refers to oral or nasogastric administration.
> > >   - User: ###END### (RULE VIOLATION)
> > >
> > > - Violated Rule: Before ending the conversation, ask the DB agent to double check that their final answer covers all goals raised. If not, request any missing parts.
> > >
> > > 2. Violation of "Missing conditions" Rule
> > >
> > > - User Instruction: Your goal is to find information related to a patient’s diagnosis. Specifically, you are interested in patient ID 10004733 and you want to know how many days have passed since the patient was diagnosed with cerebral artery occlusion during the current hospital admission. If the patient has multiple records of the same diagnosis, you want to know the number of days elapsed since the earliest diagnosis. You want diagnosis names containing "cerebral artery occlusion."
> > > - Conversation History:
> > >   - User: I'm trying to find out some information about a patient's diagnosis.
> > >   - Agent: I can help with that. What specific information are you looking for regarding a patient's diagnosis? For example, are you trying to find:
> > >     - A patient's diagnosis history?
> > >     - The most common diagnoses?
> > >     - Diagnoses associated with certain procedures or medications?
> > >     - The cost associated with a particular diagnosis?
> > >     - Something else entirely?
> > >
> > >     Please provide more details so I can narrow down my search.
> > >   - User: I'm interested in patient ID 10004733. I want to know how long ago they were diagnosed with something.
> > >   - Agent: Okay, I can help you find out how long ago patient ID 10004733 was diagnosed. Let me first check the admissiondiagnoses table to see if it contains the diagnosis date and patient ID.
> > >   - User: ###END### (RULE VIOLATION)
> > > - Violated Rule: The user must provide all detailed conditions specified in the instruction before ending the conversation.
> > >
> > > 3. Violation of "Performing Agent-like Tasks" Rule
> > > - User Instruction: Your goal is to find the number of blood culture tests performed on a specific patient. Specifically, you want to know how many culture tests were collected from venipuncture blood for patient ID 031-10300 this month.
> > > - Conversation History:
> > >   - User: I need to figure out how many blood culture tests a particular patient has had.
> > >   - Agent: To determine the number of blood culture tests, I need the patient's ID. Could you please provide the patient ID?
> > >   - User: The patient ID is 031-10300.
> > >   - (...omitted…)
> > >   - Agent: The patient with ID 031-10300 had 2 blood culture tests collected from venipuncture blood.
> > >   - User: Okay, I have the patient ID, the time frame (this month), and the collection method (venipuncture blood). Does your request include any other conditions? (RULE VIOLATION)
> > > - Violated Rule: The user acts like a DB agent or AI assistant instead of the user (e.g., writing, reviewing, or executing SQL queries, calling external APIs, or responding to the DB agent in a machine assistant way).

---

> > > > ### Comment · Reviewer_Rw7P · 2025-11-27
> > > >
> > > > Thanks for your comprehensive rebuttal, I have raised my scores accordingly.

---

> > > > > ### Author Response · Authors · 2025-11-28
> > > > >
> > > > > Thank you for your constructive feedback and the score increase. We will update the dataset soon.

---

### Official Review · Reviewer_FdSS · 2025-11-02

**Soundness:** 3
**Presentation:** 3
**Contribution:** 3
**Rating:** 6
**Confidence:** 3

**Summary:**

The authors seek to improve the benchmarking of language model agents in EHR-relevant database tasks, such as clarifying questions, solving value mismatches, and generating accurate SQL. They find that the current LLMs they test struggle with consistent success in this regard. They describe the method of benchmark creation as well as testing, including the use of LLM-based validation.

**Strengths:**

- I commend the authors for engaging with this particular topic within the Health AI field - optimizing the nuts and bolts of back-end EHR agent interaction will be very important to ensuring these tools are practically useful. Their recognition of the inherent ambiguity and clinical context-specificity of clinician questions is also important.
- The authors' introduction includes a reasonable summary of the relevant literature in the field and the purpose of their benchmark.
- The figures are clear and add to the quality of the work. The overall workflow and structure is quite clear.
- The core QA tasks are reasonable, and the level of human annotation appears overall strong (although I would love further detail on the "38 graduate-level contributors" and what was actually done).
- I greatly appreciate the depth of analysis the authors offer regarding the nature and presumed causes of errors. This strongly supports the validity of this system as a useful benchmark.
- In general, this paper appears to make a strong contribution to the broader literature.

**Weaknesses:**

- It is always difficult, when presenting such a benchmark, to know the extent to which the failures reported by the authors (e.g. the worse Pass^5 vs Pass@5) are inherent to the model vs related to flaws in the authors' implementation of that model. While this is inevitable to benchmarking, I do think it should clearly be acknowledged that these results represent a plausible floor, but not at all a ceiling for that which the authors seek to evaluate.
- Benchmarks using LLMs as user-agents have the same set of concerns, regarding the prospect for the questioner-LLM to inadvertently "tip off" the answerer-LLM, but that does not render them useless in this regard. The authors overall do a good job of highlighting these errors in their section 6, but perhaps this should also be discussed. That is, there are failures which make the test too difficult, but there also may be those which make it too easy.
- Perhaps more a direction for fture work, but as a clinician I feel that these systems can be further elevated through the use of clinical context itself to improve the questioning process, rather than just general clinical knowledge.
- I am quite concerned, however, about the authors' use of the LLM-as-judge paradigm without any human annotated gold-standard set for whether the validator actually achieves its goals. How are we to know, for example, whether these invalidations are correct, or whether this "validator" is itself discarding valid behaviors? Quis custodiet ipsos custodes? This use of unvalidated LLM-as-judge is very concerning here, and I recommend that either further validation of this approach is offered, or the section is removed.

**Questions:**

1. Who were these 38 graduate-level contributors? How were they trained? What is their background? What was their specific input?
How was the "LLM-as-judge" validator system itself validated? What were the impacts of implementing this system?

---

> ### Author Response · Authors · 2025-11-24
>
> We sincerely thank the reviewer for their positive assessment of our work, particularly for recognizing the importance of addressing inherent ambiguity and clinical context-specificity of clinician questions to develop interactive EHR agents. Please find our responses below.
>
> **[W1] It is always difficult, when presenting such a benchmark, to know the extent to which the failures reported by the authors (e.g. the worse Pass^5 vs Pass@5) are inherent to the model vs related to flaws in the authors' implementation of that model. While this is inevitable to benchmarking, I do think it should clearly be acknowledged that these results represent a plausible floor, but not at all a ceiling for that which the authors seek to evaluate.**
>
> We fully agree with the reviewer’s insightful observation that our results likely represent a plausible floor rather than a ceiling. To disentangle the intrinsic capabilities of LLMs from the complexity of agentic frameworks, we intentionally utilize a standard function-calling setup (i.e., relying on the model's native, out-of-the-box capability to select and use tools). This approach, widely adopted in frontier model evaluations (e.g., OpenAI, Gemini) and in agent benchmarks (e.g., Tau-bench), minimizes the implementation mistakes that can arise from sophisticated agentic frameworks. As suggested, we have explicitly clarified in Section 5.1 (Experimental Setup) that our results set a robust performance floor for intrinsic model reliability rather than a ceiling of optimized architectures (colored in blue).
>
> **[W2] Benchmarks using LLMs as user-agents have the same set of concerns, regarding the prospect for the questioner-LLM to inadvertently "tip off" the answerer-LLM, but that does not render them useless in this regard. The authors overall do a good job of highlighting these errors in their section 6, but perhaps this should also be discussed. That is, there are failures which make the test too difficult, but there also may be those which make it too easy.**
>
> We appreciate the reviewer’s suggestion to consider simulation artifacts from both perspectives. Regarding the specific concern of "tipping off" (making tests artificially easy), we clarify that our benchmark’s structural design inherently prevents this through two key mechanisms:
> - Strict Information Isolation: The simulated user operates solely based on the provided user instructions and behavioral rules. Crucially, the user simulator has no access to the database content, the ground truth SQL, or the answers (which are only used for simulation validation and answer evaluation). Thus, the user cannot "leak" information they do not possess.
> - Conditional Instruction Design: In AdaptQA, where user goals often change based on database findings, instructions are written in a conditional format. For instance, an instruction might read: "Find medication X. If found, that is the answer. If not found, ask for its active ingredient". Because the user does not know beforehand whether "medication X" exists in the database, they must wait for the agent's report to trigger the next step in the logic.
>
> We have revised Section 3.2 (Simulated User) and Section 4.2.3 (AdaptQA Annotation) to explicitly discuss how these design choices prevent "tipping off" during user-agent simulation.
>
> **[W3] Perhaps more a direction for fture work, but as a clinician I feel that these systems can be further elevated through the use of clinical context itself to improve the questioning process, rather than just general clinical knowledge.**
>
> We appreciate your insight from a clinical perspective and agree that incorporating richer clinical context is an important direction for future work. At the moment, however, publicly accessible EHR datasets are limited, and the strict de-identification procedures they require inevitably reduce the level of detail in patient records. Due to these constraints, the tasks in our benchmark center on patient record retrieval, statistical aggregation, and general clinical knowledge, rather than on highly specific or nuanced clinical information-seeking scenarios.
>
> To extend further regarding your suggestion, we are currently collaborating with a local tertiary hospital to apply our benchmark framework within their environment. Specifically, we aim to construct real-world clinical scenarios tailored to their private database to evaluate the agents developed in this work. However, this initiative requires navigating rigorous administrative protocols, preventing immediate completion. We therefore consider this a key objective for our long-term future research.

---

> > ### Author Response · Authors · 2025-11-24
> >
> > **[W4] I am quite concerned, however, about the authors' use of the LLM-as-judge paradigm without any human annotated gold-standard set for whether the validator actually achieves its goals. How are we to know, for example, whether these invalidations are correct, or whether this "validator" is itself discarding valid behaviors? Quis custodiet ipsos custodes? This use of unvalidated LLM-as-judge is very concerning here, and I recommend that either further validation of this approach is offered, or the section is removed.**
> >
> > We share your concern regarding the reliability of the LLM-as-a-judge validator. To address your concern, we performed a manual review of 100 randomly sampled trajectories. Our inspection identified two types of validator errors:
> > - Discarding Valid Behaviors (False Rejection): Regarding the concern that the validator might discard valid behaviors, our review shows that this occurs in 10% of cases. This is primarily due to information asymmetry: while the user does not see the ground truth SQL to prevent data leakage, the validator references it to safeguard against the user inadvertently omitting specific details required for accurate SQL generation. For instance, the validator might flag a natural user paraphrasing (e.g., asking for "hypertension" in general) as a deviation if the SQL query requires specific ICD codes (e.g., icdcode LIKE '401%'). Crucially, these false rejections only trigger a regeneration of the trajectory and do not affect the agent’s performance evaluation.
> > - Accepting Invalid Behaviors (False Acceptance): Conversely, the validator failed to catch deviations in 9% of cases. Specifically, 7% involved behavioral issues (e.g., skipping a double-check before ending the conversation or being overly polite), and 2% involved task instructions not explicitly covered by database-specific rules (e.g., being confused about the first prescription time in a single hospital admission vs the first prescription time across admissions). Importantly, these missed errors did not involve critical failures that would compromise the task's validity.
> >
> > Upon updating the dataset, we will conduct the manual trajectory evaluation again and include the resulting statistics in the revised paper.
> >
> >
> > **[Q1] Who were these 38 graduate-level contributors? How were they trained? What is their background? What was their specific input?**
> >
> > The 38 contributors were Master’s and PhD students in Computer Science and AI enrolled in a graduate-level Healthcare AI course. Rather than serving as annotators, they acted as beta testers representing the benchmark’s target audience. To prepare them for this role, we provided the full benchmark codebase, detailed documentation, a baseline agent implementation, and a real-time tutorial on how to use the benchmark, with QA sessions. Over two months, they built their own database agents to solve the tasks and reported failure cases of the simulated user through Google Forms. Their feedback helped identify issues such as the simulator revealing information too early, ending conversations prematurely, or behaving inconsistently with the user instructions due to ambiguous task phrasing. This input significantly improved the user instructions and the simulation validator. Contributor demographics and testing details are provided in Appendix A.7.
> >
> >
> > **[Q2] What were the impacts of implementing this system (LLM-as-judge)?**
> >
> > The implementation of the LLM-as-judge validator serves as a critical quality control layer. While our user simulator (Gemini-2.0-Flash) employs a self-reflection framework, our manual review revealed that it still deviated in approximately 30% of trajectories (mostly involving behavioral rules). The validator, utilizing a stronger model (Gemini-2.5-Flash), effectively filters invalid traces caused by the user, preventing the simulator's errors from confounding the agent's evaluation. Although the validator sometimes filters out valid dialogues, this conservative approach allows for a fair evaluation across simulation trials.

---

> > > ### Comment · Reviewer_FdSS · 2025-11-25
> > > **Response to Authors' Comments**
> > >
> > > These responses are satisfactory to me. I have upgraded my rating to "8" in light of the changes.

---

> > > > ### Author Response · Authors · 2025-11-25
> > > >
> > > > Thank you for the valuable discussion and the score increase. We will continue improving the dataset and upload updates by the end of the discussion period.

---

### Official Review · Reviewer_WcY6 · 2025-11-04

**Soundness:** 2
**Presentation:** 3
**Contribution:** 1
**Rating:** 2
**Confidence:** 4

**Summary:**

This paper builds a benchmark called EHR-ChatQA to test how well LLM agents can answer questions over EHR databases in a real conversation, not just single direct questions by focusing on cases where the user starts vague, clarifies over time, or changes their question mid-way — like how clinicians actually talk in real world. The benchmark requires the model to interpret the question, ask clarifying questions, write SQL, and return an answer. The contribution is mostly - a realistic test setup + showing there’s still a big gap in making LLM agents dependable for EHR querying.

**Strengths:**

I believe the authors thought well and paper is original in framing EHR QA as a conversational, iterative task rather than single-turn SQL. The quality of the benchmark construction is solid, though it could be better validated. The writing is clear, making the setup and results easy to follow. I do think the significance is moderate — it highlights an important reliability gap, but the contribution is mainly diagnostic rather than solution-oriented.

**Weaknesses:**

I believe the analysis focuses mostly on pass/fail metrics, without deeper breakdowns of failure modes or model behavior, which limits insights into improving performance. Finally, the work emphasizes problem framing rather than proposing strategies to address the reliability gap, so the contribution feels diagnostic rather than forward-moving.

**Questions:**

Couuld the authors clarify how the proposed method handles edge cases or scenarios where the stated assumptions do not hold?

---

> ### Author Response · Authors · 2025-11-24
>
> Thank you for taking the time to review our paper. Please find our responses below.
>
> **[W1] I believe the analysis focuses mostly on pass/fail metrics, without deeper breakdowns of failure modes or model behavior, which limits insights into improving performance.**
>
> We respectfully point out that Section 5.3 provides a comprehensive diagnostic analysis of failure modes beyond simple pass/fail metrics. We specifically categorized failures into "Consistent Failures" (fundamental limitations) and "Inconsistent Failures" (robustness issues) to provide granular insights:
> - Quantitative Breakdown: We identified that Value Linking Errors (46.9%) are the dominant cause of consistent failures, due to value mismatches like failing to map "Lopressor" to "metoprolol". In contrast, SQL Generation Errors (71.8%) drive inconsistent failures, revealing that models are brittle to slight variations in user phrasing.
> - Behavioral Analysis: We analyzed specific agent behaviors, such as the tendency to drop constraints (e.g., forgetting a "diabetes" filter) during multi-turn interactions, which directly points to context-tracking deficiencies rather than just SQL capability issues.
> - Actionable Roadmap: Our analysis suggests ways to develop agents in user-agent tool-interaction settings, where overcoming consistent failures requires focusing on advanced value-exploration strategies, while improving context-dependent SQL generation is key to reducing inconsistent failures (the reliability gap).
>
> These three analyses are explicitly detailed in our paper and offer valuable insights for future methodological research. If you have specific failure dimensions you would like us to add, we would be more than happy to incorporate them during the author-reviewer discussion.
>
>
> **[W2] The work emphasizes problem framing rather than proposing strategies to address the reliability gap, so the contribution feels diagnostic rather than forward-moving.**
>
> We respectfully disagree with the characterization of our work as primarily "problem framing." Beyond conceptually defining the issue, our work delivers a tangible evaluation suite and actionable directions that are needed to advance the field. We summarize our contribution as the following:
> - Curated Clinical Dataset: We provide a dataset of 350 realistic clinical tasks grounded in two diverse open-source EHR databases. We rigorously developed interactive EHR question-answering tasks in collaboration with physicians to ensure they reflect real clinical relevance and complexities.
> - Agent Evaluation Environment: We provide an environment featuring a simulated user and a dedicated validator to evaluate the agent’s ability to use tools and interact with the user to address query ambiguity and value mismatch. This serves as a reliable testbed that enables researchers to rigorously evaluate and improve the agent for interactive EHR question-answering.
> Reliability Metric Framework: We introduce a quantitative framework that measures the “reliability gap” (Pass@k - Pass^k). This metric shifts the evaluation paradigm from “optimistic performance” to “safety-critical consistency,” offering a new perspective for deploying agents in high-stakes domains.
> - Direction for Future Agent Development: We identify specific pathways for agent improvement, specifically enhancing exploration strategies to resolve value mismatches and improving context management to close the reliability gap.
>
> Collectively, these contributions move beyond problem framing by providing a realistic and practical benchmark for developing agents for interactive EHR question-answering.
>
> **[Q1] Couuld the authors clarify how the proposed method handles edge cases or scenarios where the stated assumptions do not hold?**
>
> We interpret "scenarios where assumptions do not hold" as cases where the simulated user deviates from its provided instructions or behavioral rules. Recognizing the inherent stochastic risks of LLM-based simulations, we handle these edge cases through a two-stage safety mechanism. First, we implement the simulated user with a self-reflection framework, which iteratively verifies and refines generated responses to ensure alignment with instructions before sending them to the agent. Second, we incorporate an external Simulation Validator that acts as a final gatekeeper. If the validator detects any deviation in the dialogue trajectory, the simulation is flagged as invalid and automatically re-run. To empirically verify this robustness, we conducted a manual inspection of 100 sampled trajectories. Our simulation pipeline exhibited a deviation rate of 7% in behavioral aspects (e.g., skipping a double-check before ending the conversation or being overly polite) and 2% in task instructions not explicitly covered by database-specific rules (e.g., being confused about the first prescription time in a single hospital admission vs the first prescription time across admissions).

---

### Official Review · Reviewer_6mwk · 2025-11-07

**Soundness:** 2
**Presentation:** 2
**Contribution:** 3
**Rating:** 4
**Confidence:** 3

**Summary:**

The paper proposes a new benchmark to better capture the real-world complexities of text-to-sql queries on EHR databases. They propose an environment leveraging LLMs that simulate real user interactions. By going beyond static benchmarks, the paper argues to better validate the query ambiguity and value mismatch, and their interactive resolution that are common in practice.

Disclaimer: I am not well aware of text-to-sql datasets let alone EHR-specific ones. My assessment is based only on the paper's content. It is possible that I missed any important related work.

**Strengths:**

The paper is well-written and easy to follow. The motivation for EHR text-to-sql is clear and the need for going beyond static benchmarks is also imparted well.

The paper carefully controlled contamination by changing the names of tables and columns in the database. They also sourced database and queries from real hospitals, which makes the dataset interesting.

**Weaknesses:**

W1) Details on how they simulated the two user interaction workflows: IncreQA and AdaptQA in Sections 4.2.2, 4.2.3 are not clear to me. The authors should consider elaborating or elucidating with an example.

W2) I also could not follow how they validated the final response from the interaction. Since the queries can be altered mid-way (in AdaptQA) somewhat arbitrarily, how's the evaluation done?

W3) In order to establish the reliability of dataset, it is required that the authors report some validation to ensure that the numbers reported in Table 3 are not inflated. In other words, how many of the interactions are misjudged due to failures of user-simulator, user validator, or response validator?

W4) The proposed benchmark has a heavy system component because they are proposing an environment for validation. But I did not find system setup instructions in the supplementary folder of main paper.

**Questions:**

Please comment on the questions mentioned in the Weakness section.

---

> ### Author Response · Authors · 2025-11-24
>
> **[W1] Details on how they simulated the two user interaction workflows: IncreQA and AdaptQA in Sections 4.2.2, 4.2.3 are not clear to me. The authors should consider elaborating or elucidating with an example.**
>
> We appreciate the reviewer’s suggestion to clarify the simulation workflows. In our benchmark, the simulated user's behavior is strictly governed by the annotated user instructions. Below, we elaborate on the annotation procedures and their rationale, along with specific examples.
> - Incremental Query Refinement (IncreQA): The IncreQA flow focuses on mimicking scenarios where the user progressively adds new constraints. We first curate clinically relevant questions and annotate the corresponding SQL and answers. Then, we convert the SQL into narrative user instructions by prompting an LLM to convert the semantics without missing any details, followed by manual review. For example, a SQL query like SELECT count(*) ... WHERE admitsource = 'emergency room' ... is converted into the instruction: "Your goal is to count patients... specifically those admitted from the emergency room..." We adopt this SQL-to-instruction conversion approach to avoid any ambiguities often found in typical natural language queries. For instance, a simple question like "How many emergency patients are there?" is unclear as it could refer to admission urgency (admission_type='urgent') or the admission location (admit_source='emergency room'). By deriving the instruction directly from the specific SQL query, the instructions can better reflect the intended SQL queries.
> - Adaptive Query Refinement (AdaptQA): AdaptQA focuses on scenarios involving goal adjustments during the user-agent interaction. In AdaptQA, we take different steps from IncreQA, where we first define eight query goal modification categories (Table 13), annotate the instructions, and then annotate the SQL and the answers. These eight categories represent distinct logical patterns for query modification based on clinical workflows, such as navigating medication nomenclature or identifying alternative treatments. When annotating a task for the “medication nomenclature traversal” category, we select a patient who has no records for the brand name (e.g., Lipitor) but has records for its generic name (e.g., atorvastatin). The instruction directs the user: "Find Lipitor. If not found, ask to check for its generic name." Importantly, the instruction does not explicitly state the generic name ("atorvastatin"), so the agent must identify it (e.g., using web_search) to succeed. After physician verification, we further annotate the corresponding SQL and the answers.
>
> We have revised Sections 4.2.2 and 4.2.3 to reflect these details and added Section B.1, which provides step-by-step annotation procedures.
>
>
> **[W2] I also could not follow how they validated the final response from the interaction. Since the queries can be altered mid-way (in AdaptQA) somewhat arbitrarily, how's the evaluation done?**
>
> We would like to clarify that the query changes in AdaptQA are not arbitrary, but that the interaction flow follows strictly defined conditional logic in the user instructions. As described in our response to [W1], the user simulator operates with decision-tree-like logic, where the goal adjustment depends on the agent's previous response. While the user does not know the ground truth beforehand, the evaluation is performed against the final intent reached. For instance, in the Lipitor example in our response to [W1], if the agent correctly identifies that Lipitor is missing, the user is instructed to ask the agent to check for its generic name. The agent's answer is then validated against the generic name. Conversely, if the agent fails to report the missing value and returns an incorrect answer, the user accepts this incorrect value for Lipitor, resulting in an incorrect answer.
>
> We have updated Sections 4.2.3 and Appendix A.5.2 to explicitly articulate this validation logic.

---

> > ### Author Response · Authors · 2025-11-24
> >
> > **[W3] In order to establish the reliability of dataset, it is required that the authors report some validation to ensure that the numbers reported in Table 3 are not inflated. In other words, how many of the interactions are misjudged due to failures of user-simulator, user validator, or response validator?**
> >
> > We would like to clarify that the design of our benchmark structurally prevents the user simulator from “tipping off“ the agent with information they should not possess. By strictly isolating the simulated user from the database content, ground truth SQL, and the answers, and providing only a narrative instruction and user-specific behavioral rules, we ensure that the reported scores reflect the agent’s genuine capabilities, not artifacts of information leakage. We detail these design safeguards below.
> > - IncreQA annotation: We rephrase database values into everyday language (e.g., mapping "malignant neoplasm” to "breast cancer") in the instructions. Since the user does not know the specific database schema or entry values, they cannot directly guide the agent to look for specific database schema or terms.
> > - AdaptQA annotation: As described in Section 4.2.3 (revised for clarity), the instructions for pivoting logic are designed to be purely conditional. For a task in the medication nomenclature traversal category, we select a patient lacking records for a brand name (e.g., Lipitor) but having records for its generic equivalent. The instruction guides the user to request the brand name first and, upon failure, to check for the generic name. Crucially, we intentionally omit the generic name (e.g., atorvastatin) from the instruction.  This prevents the user from revealing information that the agent must discover independently.
> > - Deterministic response validation: Our evaluation is rule-based, not model-based. As detailed in Section 3.1 and Appendix A.6, we enforce strict grading rules: executing generated SQL queries for IncreQA and verifying exact string matches within XML tags for AdaptQA.  Failures to meet these formats are counted as agent errors (these grading rules are provided to the agent), ensuring that no misjudgment or false positives occur during the scoring phase.
> >
> >
> > **[W4] The proposed benchmark has a heavy system component because they are proposing an environment for validation. But I did not find system setup instructions in the supplementary folder of main paper.**
> >
> > Thank you for raising this practical point. The original codebase we uploaded contained the full working setup (including databases, tools, and agent-user-validator orchestration) but did not include the step-by-step README file. We have added the README file to make the benchmark straightforward to set up and use.

---

### Author Response · Authors · 2025-11-24

We sincerely thank all the reviewers for their time and constructive feedback. We are encouraged by the positive assessment of our work's novelty and rigorous construction. We have carefully addressed each reviewer's comments in the individual responses below.

**Major Update Plan**: In response to the suggestion regarding the statistical robustness of the AdaptQA benchmark (raised by Reviewer Rw7P), we are actively working on expanding the dataset by constructing 16 additional high-quality tasks, which will increase the dataset size by 25% (from 64 to 80 instances). We plan to release these new samples before the end of the discussion period, and subsequently provide the updated experimental results prior to the paper decision notification.

---

> ### Author Response · Authors · 2025-12-03
>
> We have updated the dataset, and the new statistics are as follows.
>
> |           | IncreQA | AdaptQA | Total |
> | --------- | ------- | ------- | ----- |
> | MIMIC-IV★ | 145     | 40      | 185   |
> | eICU★     | 141     | 40      | 181   |
> | Total | 286     | 80      | 366   |

---

### Author Response · Authors · 2025-12-03

Dear AC,


We appreciate your time and effort in managing the review process, especially given the recent challenges the conference has faced. We would like to provide a brief summary of our discussion, focusing on the points raised by multiple reviewers, hoping that this will assist you in making a better informed decision.

**Strengths**: All four reviewers highlighted the novelty of formulating EHR question-answering (QA) in an interactive, simulated setting where agents must clarify vague user queries and use tools to complete the tasks. They also commended the rigorous benchmark construction involving clinical experts. Additionally, three out of four reviewers (Reviewers WcY6, FdSS, and Rw7P) acknowledged that our work highlights an important reliability gap (the discrepancy between optimistic and consistent success) in current LLM agents.

**Reviewer concerns and our responses**:
- Simulation Validity: Three reviewers (Reviewers 6mwk, FdSS, and Rw7P) questioned the reliability of the simulated user and the validator (e.g., potential "tipping off" or score inflation). We clarified that the simulated user is structurally isolated from the database content to prevent information leakage. We also presented a manual inspection of 100 trajectories and provided concrete examples to illustrate how the validator successfully detects and filters rule violations. This demonstrated that the validator effectively enforces the simulation rules without inflating scores. Reviewers FdSS and Rw7P accepted our explanation. We have not received any response from Reviewer 6mwk.
- Failure Analysis: Two reviewers (Reviewers WcY6 and Rw7P) suggested deeper diagnostic insights in our experiments. We respectfully pointed out that Section 5.3 of our original submission already provided a comprehensive breakdown of these failure modes. We clarified that while consistent failures often stem from value linking issues (e.g., failing to map a brand name like 'Lopressor' to its generic name 'metoprolol'), the reliability gap is primarily driven by context-dependent SQL generation errors (e.g., losing track of previous dialogue context). Reviewer Rw7P accepted our clarification. We have not received any response from Reviewer WcY6.
- Dataset Size: Reviewer Rw7P noted that the AdaptQA dataset was small (64 instances) and asked about our plans to expand its size. We committed to adding 16 new task instances for AdaptQA to increase the dataset size by 25 percent. Reviewer Rw7P was satisfied with our plan.

**Current Status**: Based on these clarifications and updates, Reviewer FdSS raised their score from 6 to 8, and Reviewer Rw7P raised their score from 4 to 6. We have not received any response from Reviewer 6mwk (Score: 4) or Reviewer WcY6 (Score: 2). We leave the final decision to your discretion.


Best,

Authors

---

### Meta-Review · Area_Chair_v9Tk · 2025-12-06

**Summary:**

The paper introduces EHR-ChatQA, a novel benchmark designed to evaluate LLM agents in interactive Electronic Health Record (EHR) question-answering scenarios. Unlike static text-to-SQL benchmarks, this work focuses on the agent's ability to handle query ambiguity, resolve value mismatches, and utilize tools through multi-turn interactions. The authors introduce a "reliability gap" metric to quantify the difference between optimistic performance (Pass@k) and consistent reliability (Pass^k).

The decision to accept is based on the paper's clear motivation, the rigorous construction of the simulation environment, and the significant improvements made during the rebuttal period. While initial scores were mixed, the active engagement from authors—including expanding the dataset size and providing manual audits of their validation system—successfully persuaded the engaged reviewers. The remaining negative sentiment stems primarily from a preference for modeling contributions over benchmarking contributions, which does not diminish the value of this work to the community.

**Reviewer Concerns:**

Addressed by Rebuttal:

1. Simulation Validity and "Tipping Off": Reviewers FdSS and 6mwk raised concerns that the simulated user might inadvertently leak ground truth information to the agent. The authors clarified the structural isolation of the user simulator (which does not have access to the DB content) and the conditional logic used in AdaptQA. Reviewer FdSS explicitly accepted this explanation.

2. LLM-as-a-Judge Reliability: Reviewer FdSS questioned the reliability of using an LLM validator without human verification. The authors performed a manual inspection of 100 trajectories, reporting specific error rates (10% false rejection, 9% false acceptance) and explaining that the validator errs on the side of caution. This satisfied Reviewer FdSS.

3. Dataset Size: Reviewer Rw7P noted that the AdaptQA subset (64 instances) was small. The authors committed to and executed a plan to add 16 new high-quality instances (a 25% increase) and updated the statistics. Reviewer Rw7P raised their score based on this responsiveness.

Outstanding / Unresolved:

1. Contribution: Reviewer WcY6 maintained a lower score, criticizing the work for being "diagnostic rather than solution-oriented." The authors argued that for a benchmark paper, establishing a rigorous evaluation environment and identifying the "reliability gap" is the primary contribution. I agree with the authors; this criticism seems to reflect a preference for algorithmic novelty rather than a flaw in the benchmark itself.

Reviewer 6mwk's Final Confirmation: Reviewer 6mwk did not respond to the rebuttal despite the authors addressing the specific technical questions regarding simulation details. However, the provided clarifications appear robust.

**Reviewer Scores:**

Based on the authors' clarifications and updates, Reviewer FdSS raised their score from 6 to 8, and Reviewer Rw7P raised their score from 4 to 6. We have not received any response from Reviewer 6mwk (Score: 4) or Reviewer WcY6 (Score: 2). We leave the final decision to your discretion.

---

### Decision · Program_Chairs · 2026-01-26

Accept (Poster)